# Active Source Rock Depth Limit and its Controlling on the Formation and Occurrence of Fossil Fuel Resources

Xiongqi Pang[1,2] *, Chengzao Jia[1,3], Kun Zhang[2,4] **, Maowen Li[5], Youwei Wang[2,6], Junwen Peng[7], Boyuan Li[2], and Junqing Chen[1,2]

[1]State Key Laboratory of Petroleum Resources and Prospecting, China University of Petroleum, Beijing, 102249, China
[2]College of Geosciences, China University of Petroleum, Beijing, 102249, China
[3]Research Institute of Petroleum Exploration and Development, PetroChina, Beijing, 100083, China
[4]Department of Earth Sciences, University College London, Gower Street, London WC1E 6BT, UK
[5]State Key Laboratory of Shale Oil and Gas Enrichment Mechanisms and Effective Development, SINOPEC Exploration and Production Research Institute, Beijing, 100083, China
[6] Department of Geosciences and Engineering, Delft University of Technology, Stevinweg 1, 2628 CN Delft, Netherlands
[7]Bureau of Economic Geology, University of Texas at Austin, Austin, TX 78713, USA

*Correspondence to*: Xiongqi Pang (pangxq@cup.edu.cn); Kun Zhang (zhangk.cupb@gmail.com)

**Abstract.** Fossil fuel resources are invaluable to economic growth and social development. Understanding the formation and distribution of fossil fuel resources is critical to the search and exploration of them. Until now, the vertical distribution depth of fossil fuel resources has not been confirmed due to different understandings of their origins and the substantial variation of reservoir depths from basin to basin. Geological and geochemical data of 13,634 source rock samples from 1,286 exploration wells in six representative petroliferous basins were examined to identify the maximum burial depth of active source rocks in each basin, which is named in this study as the active source rock depth limit (ASDL). Beyond ASDL, e source rocks no longer generate or expel hydrocarbons and become inactive. Therefore, ASDL also sets the maximum depth for fossil fuel resources. The ASDLs of basins over the world are found to range from 3,000 m to 16,000 m, while the thermal maturities (Ro) of source rocks at the ASDLs are almost the same, with Ro≈3.5±0.5%. The Ro of 3.5% can be regarded as a general criterion to identify ASDLs. High heat flow and more oil-prone kerogen are associated with shallow ASDLs. In addition, tectonic uplift of source rocks can significantly affect ASDLs. 21.6 billion tons of reserves in six representative basins in China and 52,926 documented oil and gas reservoirs in 1,186 basins over the world are all located above ASDLs, demonstrating the universal presence of ASDLs in petroliferous basins and their control on the vertical distribution of fossil fuel resources. The data used in this study are deposited in the repository of the PANGAEA database: https://doi.pangaea.de/10.1594/PANGAEA.900865 (Pang et al., 2019).

**Keywords**: Fossil fuels; Nature energy; Conventional and unconventional hydrocarbons; Sedimentary basin; Active source rock depth limit

# 1 Introduction

Fossil fuel resources, including coal, conventional and unconventional hydrocarbons, account for 85.5% and 86.9% of the primary energy consumption in 2016 of the world and China (B.P. Global, 2017), respectively. Because of their indispensable role in the world economy, a lot of research have been done on fossil fuels in the past few decades, including
characterizing and explaining their spatial distribution in various types of sedimentary basins (Tissot & Welte, 1978; Wang et al., 1997; Gautier et al., 2009) and their temporal distribution through the past 1.6 billion years in the geological history (Wang et al., 2016). However, the vertical distribution of fossil fuel resources, especially the maximum preservation depth, is still under debate because of different understandings of the fossil fuel resource origins and the great variations of depths from basin to basin (Kennedy et al., 2002; Peters et al., 2005; Pang et al., 2015).

As global demand for energy keeps rising, fossil fuel exploration is rapidly expanding to more challenging and deep regions of the Earth (Dyman et al., 2002). Currently, the deepest commercial hydrocarbon reservoir worldwide is located in the basin of Mexico Gulf with a depth of 11,945 m (including water depth) (Transocean, 2009). In China, deep (> 4,500 m) and ultra-deep (> 6,000 m) oil and gas reservoirs are mainly found in the Tarim Basin, where the amount of deep oil and gas reserve is estimated to account for more than 90% of the total proved reserves (Pang et al., 2015). In order to boost oil and gas
supply to support fast economic growth, China initiated research programs developing 10,000-m-scientific drilling rigs, and funded the National Basic Research Program (973 Program) to better understand hydrocarbon accumulations deep in basins (Jia et al., 2016). One major challenge for deep oil and gas exploration comes from the significant variation of reservoir depths in different basins and the uncertainty it poses to oil and gas resource assessment. In some basins, dry layers, target strata containing no oil or gas, are prevalent at a depth of 4,500 m or less, whereas in some other basins, the maximum burial depth
for oil and gas accumulation is predicted to more than 10,000 m. To date, the maximum depth to which fossil fuels can be formed and preserved in the Earth's crust remains unresolved. Some researchers support the abiogenic petroleum origin and believe that the maximum depth of hydrocarbon occurrence is much deeper than the maximum depth of petroliferous basins (Gold, 1993; Kenney et al., 2002). However, more and more researchers believe that oil and gas are of biogenic origin and suggest that the maximum depth of oil and gas reservoirs is critically controlled by the depth of active source rocks which
generate and expel oil and gas in sedimentary basins (Tissot & Welte, 1978; Durand, 1980; Hunt, 1996).

To solve this mystery and to understand hydrocarbon generation and accumulation processes, this study selected six representative petroliferous basins in China, which have the largest areas, the largest proved oil and gas reserves and the highest exploration degrees  (Fig. 1, Table 1), to identify the maximum depth of fossil fuel resources in each basin and investigate factors leading to the variation of the maximum depth from one basin to another. This study did not take the abiogenic
petroleum origin into account for the reason that the genetic relationship between petroleum and organic matter in source rocks are proved and widely accepted (Magoon & Dow, 1994; Peters et al., 2005). Besides, no commercial petroleum reservoirs of abiogenic origins have been discovered to date (Kenney et al., 2002; Glasby, 2006; Höök et al., 2010; Selley & Sonnenberg, 2014). In this study, geological and geochemical data of 13,634 source rock samples from 1,286 exploration wells in six basins

were examined. The maximum depth for the formation and occurrence of fossil fuel resources in these basins were determined. Major geological factors influencing the maximum depths of active source rocks were analysed and their controlling on the distribution of fossil fuel resources were discussed.

## 2 Materials and Methods

### 2.1 Study sites and data collection

We conducted this study regarding ASDLs in six representative basins in China, including the Songliao Basin and the Bohai Bay Basin in Eastern China, the Sichuan Basin and the Ordos Basin in Central China, and the Tarim Basin and the Junggar Basin in Western China. For each basin, we utilized at least four different indicators detailed in the Section 2.2 to determine the ASDL. The data were obtained from PetroChina and Sinopec and are available through the PANGAEA database:

https://doi.pangaea.de/10.1594/PANGAEA.900865 (Pang et al., 2019). We also investigated the relationships of ASDLs and the distributions of 52,926 reservoirs in 1,186 basins over the world according to the database of IHS (2010) to verify its universality.

### 2.2 Characterization of ASDLs

Active source rocks are sedimentary rocks rich in organic matter and capable of generating hydrocarbons. In the evolution

history of a basin that spans over millions of years, source rocks are activated and producing hydrocarbons at certain conditions, such as the generally regarded threshold temperature of 60 ˚C (Tissot & Welte, 1978; Peters, 1994). With further increase of burial depth of the source rocks, the potential amount of hydrocarbons that can be produced and expelled from the source rocks decreases and eventually approaches zero. The Active Source Rock Depth Limit (ASDL) is defined as the maximum burial depth of active source rocks beyond which the source rocks no longer generate or expel hydrocarbons and become inactive. In

addition to the burial depth, ASDL can also be characterized by other physical parameters of source rocks, such as the thermal maturity.

The potential amount of hydrocarbons that can be further generated from a source rock sample cannot be directly measured, but can be evaluated based on many experimentally measurable parameters, such as the atomic number ratios of hydrogen to carbon (H/C) and oxygen to carbon (O/C) of the remaining organic matter in the sample. The generation of oil

and gas from organic matter is the process of condensation of the aromatic nuclei that enriches carbon by deoxygenation and dehydrogenation. The process can be experimentally studied by measuring the decrease in the H/C and O/C ratios (Tissot et al., 1974). In theory, organic matter in source rocks eventually evolves to graphite with increasing thermal maturity with the H/C and O/C ratios drop to zero. This indicates that the active source rocks no longer produce hydrocarbons and thus reach the ASDL.

Rock-Eval pyrolysis parameters can also be utilized to identify the ASDL such as the hydrocarbon generation potential index ("$S_1$ + $S_2$"/TOC). "$S_1$" is the amount of hydrocarbons released from a source rock sample when it is heated from room

temperature to 300 °C, and "$S_2$" is the amount released from 300 °C to 600 °C. TOC is the measured total organic carbon in the source rock sample (Espitalie et al., 1985). The concept of the hydrocarbon generation potential index was proposed by Zhou and Pang (2002). Pang et al. (2005) used the index to measure the quantity of hydrocarbons that can be generated from a single unit weight of organic carbon. The index generally increases with increasing burial depth when the thermal maturity is low and then decreases with increasingly higher burial depth or thermal maturity. The turning point of hydrocarbon generation potential index corresponds to the hydrocarbon expulsion threshold (HET) which was proposed by Pang et al. (1997). HET represents that hydrocarbons start migrating out of source rocks to surrounding reservoirs. As the expulsion continues, the hydrocarbon generation potential index gradually decreases. When the index approaches zero, source rocks can no longer expel hydrocarbons and reach the ASDL. Along with the evolution of hydrocarbon generation potential index, the hydrocarbon expulsion ratio (Qe), hydrocarbon expulsion rate (Ve) and hydrocarbon expulsion efficiency (Ke) of the source rocks also evolve with thermal maturity. Qe represents the amounts of hydrocarbons expelled from a unit weight of organic carbon. Ve represents the hydrocarbons expelled from a unit weight of organic carbon when the burial depth increases by 100 m. Ke represents the ratio of the cumulative amount of hydrocarbons expelled from source rocks to the cumulative amounts of hydrocarbons generated. When source rocks reach ASDL, Qe and Ke approach the constant values and Ve approaches the value of zero.

Hydrocarbon generation is the transformation of original organic matter, also referred to as kerogen, to transitional compounds and finally to hydrocarbons (Behar, et al., 2006). When the amount of transitional compounds or residual hydrocarbons ("$S_1$" or "A") decrease to zero, the hydrocarbon generation potential is also exhausted. Experimentally, "A" is the amount of hydrocarbons extracted by a chloroform solution from a source rock sample. Because some non-hydrocarbon compounds are also extracted, "A" is generally larger than "$S_1$". The residual hydrocarbon content index ("$S_1$"/TOC or "A"/TOC), which represents the quantity of hydrocarbons retained per unit weight of organic carbon, can therefore be used to indicate the ASDL. Previous studies (Zhou and Pang, 2002; Pang et al., 2005) indicate that source rocks reach HET when the residual hydrocarbon content index reaches its maximum value. After that, the index began to decrease. The source rocks finally evolve to pass the ASDL to become inactive when the residual hydrocarbon content index decreases to a minimum value. In summary, the parameters listed in the section, including H/C, O/C, "$S_1 + S_2$"/TOC, "$S_1$"/TOC, "A"/TOC, Ve and Ke, all trend as a function of source rock burial depth (D) or thermal maturity (Ro). The ASDL can thus be represented as the critical values of D or Ro when the indexes of H/C, O/C, "$S_1 + S_2$"/TOC, "$S_1$"/TOC, "A"/TOC, and Ve approach zero, or when Ke approaches a constant value.

## 3 Results and Discussions

### 3.1 ASDLs in the Six Representative Basins

The ASDLs of the six representative basins were characterized. The Junggar Basin located in Western China is used as an example to illustrate the process of characterization (Fig. 2). The same methods were applied to study the other five basins,

and the results are shown in Figs. S1–S5 and Table 2. The hydrocarbon formation and accumulation in the Junggar Basin are mainly controlled by the Permian petroleum system (Wang et al., 2001). Previous geochemical and sedimentological data demonstrate that the source rocks are mainly Permian shales and that the main reservoirs are the clastic rocks in the Permian, the Triassic, and the Jurassic Formations capped by the Upper Triassic, the Lower Jurassic, and the Lower Cretaceous mudstones, respectively (Cao et al., 2005). A few Carboniferous volcanic reservoirs are found distributed in structural highs near fault zones and unconformities, the hydrocarbons in these reservoirs are also primarily derived from the Permian shales (Chen et al., 2016; Wang et al., 2018). According to the analyses of fluid inclusions and basin modelling, the Permian source rocks started generating hydrocarbons since the Middle-Late Permian due to a rifting process-related high heat flow, and the main hydrocarbon accumulation period spanned from the Triassic to the Paleogene for the whole basin (Wang et al., 2001; Cao et al., 2005). Petroleum systems in the other five basins were studied by other researchers (Zhou and Littke, 1999; Xiao et al., 2005; Wu et al., 2008; Ping et al., 2017; Zhu et al., 2018).

In the Method Section, we provided theoretical threshold values of different geochemical parameters or indexes to indicate ASDL. In practice, envelope lines enclosing all sample data points are utilized to show the overall trends of how these parameters change with increasing burial depth or thermal maturity. The interceptions of the envelope lines with these threshold values represent source rocks reaching ASDLs. This envelop method has been widely and successfully employed in a variety of basins in China, and numerous studies containing different geochemical data and mathematical models have been published (Zhou and Pang, 2002; Pang et al., 2004; Jiang et al., 2016; Peng et al., 2018). It is found the profiles of hydrocarbon generation potential index, Ve, and residual hydrocarbons are overall bell-shaped, though details can vary depending on the source rock types (e.g. different lithologies and organic matter types). On the other hand, some uncertainties may exist in the envelope method due to the lack of data from ultra-deep wells. In this case, the ASDLs can be identified by extrapolating the profiles according to the variation trends established based on the available data at different burial depths or thermal maturities. The envelope lines employed in this study are guided by well-established models and trends derived from actual geochemical data.

Figure. 2a shows H/C ratios of source rock samples from the Permian shales plotted against burial depth. The average H/C ratio decreases sharply at a depth of about 6,000 m, beyond which there are no samples with H/C ratios greater than 1.5. The intercept of the dashed line on the vertical axis marks the ASDL, which corresponds to D $\approx$ 8,350 m and Ro $\approx$ 3.0%. Figure. 2b shows the variation of residual hydrocarbon amounts in source rock samples, represented respectively by "A"/TOC or "$S_1$"/TOC, with burial depth. Initially, both the mean and the variance of the residual amounts increase with depth, because hydrocarbons are generated but not yet expelled out of the source rocks. The mean reached the maximum at the depth of 3,500 to 4,000 m or at Ro$\approx$1.0%, which is the hydrocarbon expulsion threshold. With further increase of depth, the amount of residual hydrocarbon starts decreasing, and eventually reaches zero at a depth of 7,850–7,960 m and a corresponding Ro of 3.0%, indicating the ASDL. Figure. 2c shows the change of hydrocarbon generation potential index, ("$S_1$ + $S_2$")/TOC, hydrocarbon expulsion ratio (Qe), hydrocarbon expulsion rate (Ve) and hydrocarbon expulsion efficiency (Ke) of the source rock samples with increasing burial depth. These results indicate an ASDL of 8,200 m with Ro of 3.0%, in good agreement with the ASDL

values obtained in Fig. 2a and 2b. In addition, the HET is determined to be D of 3,000 m and Ro of 0.9%, and the hydrocarbon expulsion peak occurs at D of 4,500 m and Ro of 1.3%.

According to the ASDLs identified for the six representative basins (Table 2), three general conclusions on ASDL can be drawn. First, for the same basin, the ASDLs derived from the six geochemical indexes are the same or very close in values. For the Junggar Basin, the derived ASDLs vary from 7,850 m to 8,450 m with an average value of 8,168 m and a deviation of 7.6%. Second, ASDLs in different basins can be very different. ASDLs of the six basins range between 5,280 m and 9,300 m with an average value of 7,094 m and a deviation of >76.1%. Third, for all the ASDLs of the six basins, the corresponding thermal maturities (Ro) have much smaller variation than the depths. Ro values vary from 3.0% in the Junggar Basin to 4.0% in the Songliao Basin, with an average of 3.5% among the six basins and a deviation of 33.3% much smaller than the 76.1% deviation of the depths. This implies that ASDL is mainly controlled by the thermal maturity of source rocks. The average thermal maturity level of 3.5% derived in this study can be regarded as the identification criterion for ASDL in general geological settings.

## 3.2 Major factors controlling ASDLs and their effects

### 3.2.1 Organic Matter Type

Original organic matter (or kerogen) in source rocks is generally classified into three types based on its origin (Peters, 1994; Tissot et al., 1974). The three types have different organic element compositions and different pyrolytic parameters, and therefore have different hydrocarbon generation potentials. The hydrocarbon generation potential indexes of different type source rock samples from the representative basins are plotted in Fig. 3. The dashed curves enveloping all the sample data points indicate the varying trends of hydrocarbon generation potential of source rocks with different organic matter types. The trends with thermal maturity (Ro) are very similar for all three organic matter types: the index first increases with increasing Ro and then decreases after source rocks reach HET. Source rocks with type I (oil-prone), type II, and type III (gas-prone) kerogens reach ASDLs at Ro of 3.0%, 3.5% and 4.0%, respectively. This indicates that oil-prone source rocks are more likely to reach the ASDL and stop generating and expelling hydrocarbons at a shallower burial depth than the other two types of source rocks under similar geological conditions.

### 3.2.2 Heat Flow and Geothermal Gradient

ASDLs are shallow in petroliferous basins with high heat flow and high geothermal gradient. The ASDLs in the six basins span from 5,400 m to 9,300 m as determined from hydrocarbon generation potential index (Fig. 4). The basins in Western China have low heat flow and low geothermal gradient (1.5–2.8 °C/100 m) and thus have the deepest ASDLs ranging from 8,200 m to 9,300 m. The basins in Eastern China are of high heat flow and high geothermal gradient (3.0–4.2 °C/100 m) and then have the shallowest ASDLs, ranging from 5,400 m to 5,900 m. The basins in Central China are moderate in terms of heat

flow and geothermal gradient, and the depths of ASDLs vary from 6,600 m to 7,700 m. In addition, the source rock burial depths corresponding to HETs vary similarly: high heat flow and high geothermal gradient lead to shallow HETs (Fig. 4).

### 3.2.3 Tectonic movement, stratigraphic age and other factors

ASDL is also influenced by other two important factors, i.e. tectonic uplift and stratigraphic age of source rocks. As previous stated, ASDL is better characterized by thermal maturity than by depth, and Ro=3.5% is regarded as general threshold for ASDL in common geological settings. However, the corresponding depth of ASDL for different source rock layers is highly variable. Due to the irreversible nature of vitrinite reflectance (Hayes, 1991; Peters, 2018), the depth of ASDL for source rocks that were historically uplifted after reaching the original ASDL is relatively shallower compared with younger source rocks that were not uplifted. The Sichuan Basin that experienced several stages of tectonic uplift in the geological history epitomizes the influence of these factors on ASDL. For example, the Ro of the upper Triassic source rocks is about 1.0% at the depth of ~2000 m in the Southern Sichuan basin (Zhu et al., 2016). At the same burial depth, however, the Ro of the lower Triassic source rocks can reach 2.0% (Zhu et al., 2016). Therefore, tectonic uplift and stratigraphic age of source rocks can have a significant effect on the corresponding depth of ASDL.

In addition to the mentioned four main factors, deep thermal fluids and overpressure retardation may also affect ASDL (McTavish, 1998; Hao et al., 2007; Fetter et al., 2019). Although it is not the scope of our study to investigate every single influence factor of ASDL in detail, we present a brief introduction to the possible consequence of these factors. Deep thermal fluids provide both fluids and a thermal source and can facilitate the maturation of organic matter. On the one hand, the conduction of thermal fluids through rocks and faults brings thermal energy to source rocks and promotes source rock maturation and hydrocarbon generation (Rullkötter et al., 1988). On the other hand, the $H_2$ brought by the deep fluids can considerably improve the hydrocarbon generation rate through the kerogen hydrogenation process (Zhu et al., 2017). Consequently, compared with unaffected source rocks, source rocks influenced by deep thermal fluids may have shallower ASDLs. In terms of overpressure retardation, an overpressure on source rocks can retard the thermal evolution of hydrogen-rich kerogen and/or the thermal cracking of hydrocarbons (McTavish, 1998; Hao et al., 2007). As a result, source rocks influenced by overpressure retardation have deeper ASDLs. It is worth noting that the thermal maturity corresponding to ASDL remains the same, no matter the ASDL becomes deeper or shallower. Namely, a source rock will reach ASDL when its Ro increases to 3.5% ± 0.5% and its hydrocarbon generation potential is depleted. Therefore, we argue that the thermal maturity of organic matter is more suitable to characterize ASDL than depth.

### 3.3 Quantitative relationship between ASDL and heat flow and organic matter type

According to the analysis in the previous section, heat flow and organic matter type act as the two main factors controlling ASDLs. In this section, a quantitative relationship is further established by statistics using the software Origin 2019. We first analysed the depths of ASDLs as a function of heat flow with a linear model. The ASDL for each basin is the average depth obtained from various geochemical indicators. The heat flow utilized in the model is the average of present heat flow values

measured at different locations in each basin (Table 1). A strong negative correlation is observed between the ASDLs and the present heat flows with a coefficient larger than 0.9 (Fig. 5a), indicating that high heat flow very likely leads to a shallow ASDL. Considering that the heat flow values of a sedimentary basin vary with geologic time, the average heat flow since the deposition of source rocks was further employed. As shown in Fig.5a, the ASDLs also present an obvious negative correlation

with average paleo-heat flows. This implies that the paleo and present heat flows both contribute to the thermal maturation of source rocks and therefore play an important role in controlling the ASDLs. We mainly utilize the present heat flow values in the following discussion, mainly because the correlation (R=0.90) between ASDL and present heat flow is much higher than that (R=0.77) between ASDL and the average value of paleo-heat flow. It is also observed that the maximum buried depth of oil-bearing targets in most basins is mainly corresponding to the maximum temperature under the current heat flow. ASDLs

for basins of different current heat flows range between 3,000 m and 16,000 m. Generally, ASDLs are less than 6,000 m in basins with high heat flow (>70 mW/m$^2$), and are greater than 9,000 m in basins with low heat flow (<40 mW/m$^2$). Given that ASDL is also influenced by organic matter type, we further analysed the effects of organic matter type on ASDL by adding the hydrogen index (HI), an indicator of organic matter type, to the linear model. HI is a quantitative proxy for the characterization of kerogen types, and is easily obtained through Rock-Eval analysis. Numerous studies on source rock

evaluation from the scientific community have proven the reliability of HI. Furthermore, HI has been widely chosen as the indicator of kerogen type in professional software, such as PetroMod, which is often used by the industrial community. To quantify the influence of organic matter types on ASDLs, the hydrogen index values of 600 mg HC/g TOC, 450 mg HC/g TOC, 525 mg HC/g TOC, 250 mg HC/g TOC and 125 mg HC/g TOC are assigned to type I, I–II, II, II–III and III kerogens, respectively. The following equation is then deduced:

$$ASDL = 16448 - 3.61 * HI - 139.46 * HF \qquad (1)$$

where ASDL is the active source rock depth limit with a unit of meter; HI is the hydrogen index value of the major source rocks in a basin, in the unit of mg HC/g TOC; HF is the present average heat flow value of a basin, in the unit of mW/m$^2$.

Although Eq. (1) shows a high correlation coefficient of 0.96 (Fig. 5b), this equation, instead of being utilized to precisely predict the ASDL of a basin, is only presented to confirm the existence of a relationship among the ASDL, heat flow and

organic matter type because of the following reasons. First, the variation of organic matter types in our study is relatively small (Table 1), and therefore, the hydrogen index values utilized to deduce Eq. (1) show small variations, which can bring uncertainties to some extent. Second, as mentioned in the above section, the ASDL is not only influenced by the heat flow and organic matter type, but also influenced by the stratigraphic age and tectonic uplift. The Eq. (1), having not included all the 4 major factors, is therefore not sufficient to predict the precise ASDL of a basin. To set up a model with four independent

variables, however, is difficult and impossible by our database of 6 basins. Construction of a complete and precise model or equation needs help from the scientific community to enrich the database. We suggest that basin modelling and other integrated analysis methods should be applied if readers want to predict the depth of ASDL in a basin without enough geological and

geochemical data. Quantitative relationship indicated in Eq. (1) provides preliminary insights into the geological basis and boundary condition for the prediction of fossil fuel distribution in the basins and helps the evaluation of hydrocarbon potential.

**3.4 ASDL controlling the vertical distribution of fossil fuel resources**

Fossil fuel resources formed from organic matter in the course of millions of years. They are currently the primary energy
sources in the world, and can be utilized in many different industries. Oil and gas are the products during the evolution of organic matter, while coal is the residue of organic matter. ASDL is the critical condition or the dynamical boundary at which oil and gas expulsion ends. It controls the formation and distribution of all economical hydrocarbon reservoirs. Once the burial depth of organic matter exceeds the ASDL, the hydrocarbons are no longer generated from the source rocks, and the coal evolves to graphite losing their industrial value as fuel. Theoretically, ASDL represents the maximum depth of the formation
and distribution of fossil fuels. According to Fig. 6, approximately 97.7% of coal resources in China and 97.3% of recoverable coal reserves over the world are distributed above ASDLs corresponding to Ro of 4.0% (CCRR, 1996; CNACG, 2016; Conti et al., 2016). Therefore, ASDL represents the maximum depth of hydrocarbon reservoir distribution, including oil, gas and coal.

This study also analysed the drilling results for 116,489 samples of target layers from 4,978 exploration wells of the six
basins in China (Fig. 7). The data show that all the reservoirs in the six basins distributed above the ASDLs, reflecting the control of ASDL on the formation and distribution of hydrocarbon reservoirs. The probability of drilling commercial oil and gas reservoirs decreases with increasing burial depth, whereas the probability of drilling dry layers increases. At some depth, the probability of drilling oil or gas reservoirs decreases to zero, and this depth is regarded as the Hydrocarbon Accumulation Depth Limit (HADL). Similar to ASDL, HADL is also influenced by many factors such as the hydrocarbon phases, the
geothermal field, the strata age and lithology of the reservoir, and will be discussed in other papers. Here, we just focus on the relationship between HADL and ASDL. The HADLs of the six basins are marked in Fig. 7 as yellow dots and connected by a dashed red line. The ASDLs deduced from ("S$_1$ + S$_2$")/TOC (Table 2) are also marked in Fig. 7 and connected with a solid blue line. Meanwhile, according to the vertical distribution characteristics of proved hydrocarbon reserves, it is observed that all proved hydrocarbon reserves in the six representative basins are controlled by the HADLs which is above the ASDLs (Fig.
7; Fig. S6). This means that the HADL in a basin is controlled by its ASDL and should always be above the ASDL. The currently discovered natural gas hydrate over the world are also distributed in fields with active source rocks (Dai et al., 2017). We further extended the research to 52,926 reservoirs in 1,186 basins over the world recorded in IHS (2010). HADL for each basin was derived from the actual reservoir depth data in IHS (2010) using the same way as described in the previous paragraph (Fig. 7) and the results are shown in Fig. 8. ASDL for each basin is assumed to be at Ro of 3.5%, and the corresponding depth
is obtained from the documented heat flow of that basin. We found that the HADLs (represented as depth) are universally above the ASDLs for all the basins.

Hydrocarbons are generally classified in two big categories as natural gas and liquid petroleum, which have distinct physical properties. By definition, ASDL marks the end of generation of any hydrocarbon from source rocks, but this concept

can be modified to incorporate the two types of hydrocarbons. Therefore, two ASDLs are introduced, including ASDLg for gas and ASDLo for oil. ASDLo indicates source rocks can no longer generate oil, and is named oil supplying depth limit. ASDLg indicates source rocks can no longer generate gas, and is named gas supplying depth limit. Hydrocarbons generated and exposed from source rocks of low thermal maturities are mainly liquid oil and gaseous hydrocarbons. The gaseous hydrocarbons become the dominant components with nearly no liquid oil when the thermal maturity is high. Therefore, theoretically speaking, the burial depth and thermal maturity corresponding to ASDLo should be lower than that of ASDLg. To investigate the ASDLs for different fluids, the high temperature (room temperature to 600 °C) and high pressure (50 MPa) pyrolysis simulation experiments were conducted on immature or low maturity kerogens sampled from Junggar Basin in a closed system. According to the experiment results, source rocks reach ASDLo at Ro of about 2.0% (Fig. 9), and the same source rocks reach ASDLg at Ro of 3.0 to 4.0%.

Besides, Pang et al. (2005) proposed the concept of hydrocarbon expulsion threshold (HET), which marks the starting point of source rocks expelling hydrocarbons at a certain depth. The HET, ASDLo and ASDLg divide a basin into three regions in the vertical direction, and they control the types of hydrocarbon reservoirs and their distributions (Fig. 10). The upper field (blue area in Fig. 10) is favourable for hydrocarbons migrating upward to form conventional reservoirs in traps, and the source rocks in this field do not yet expel hydrocarbons. The middle field (pink area in Fig. 10) is favourable for source rocks to generate, expel and retain hydrocarbons to form various kinds of oil/gas reservoirs, and the source rocks in this field supply hydrocarbons that may migrate into the upper area. The lower area (yellow area in Fig. 10) is favourable for source rocks to generate, expel and retain natural gas to form mainly unconventional resources. Figure. 10 includes a series of low-heat-flow to high-heat-flow basins in the world and illustrates the effect of heat flow on the distribution of HETs and ASDLs. The characteristics of hydrocarbon generation and reservoir distribution differ among these basins due to their different geological conditions and tectonic settings.

## 4. Conclusions

(1) ASDL is the maximum burial depth for source rocks to generate and expel hydrocarbons from geothermal cracking of kerogen. ASDL marks the depletion of hydrocarbon generation potentials of source rocks, and it commonly exists in petroliferous basins. We found the thermal maturity of 3.5% can be regarded as the identification criterion of ASDL in general geological conditions.

(2) The ASDLs of all basins over the world vary from 3,000 m to 16,000 m, and this variation is mainly caused by heat flows, kerogen type, age of source rock strata, and tectonic movement. The ASDL of a basin is deep when the basin's heat flow is low or the source rock kerogen is oil-prone. Tectonic uplift of source rock strata can significantly reduce the ASDL.

(3) All types of fossil fuel resources, including coal, conventional and unconventional oil and gas are formed and distributed above the ASDLs. A basin can be vertically divided into three fields by the HET, the oil supply limit and the gas supply limit. The three fields are favourable for different types of reservoirs.

## Data availability

The datasets can be accessed through https://doi.pangaea.de/10.1594/PANGAEA.900865.

## Author contributions

Xiongqi Pang proposed the concept of ASDL, designed the study and led the writing of the manuscript in close collaboration with Kun Zhang and Junqing Chen. The data used in this study was collected by Youwei Wang and Boyuan Li, respectively. Chengzao Jia helped collect the data and explained the significance of ASDL. Kun Zhang investigated the influence of geothermal gradient on ASDL. Maowen Li studied the influence of organic matter type on ASDL. Junwen Peng illustrated the mechanism of depletion of hydrocarbon generation potential. All authors reviewed and approved the manuscript.

## Competing interests

The authors declare that they have no conflict of interest.

## Acknowledgements

This study is supported by the National Basic Research Program (973 Program) of China (2006CB202300; 2011CB2011) and the Application Foundation Research Program of PetroChina, Sinopec, and CNOOC. We also thank the Tarim Oilfield Company, Xinjiang Oilfield Company, Liaohe Oilfield Company, Southwest Oilfield Company, Daqing Oilfield Company of PetroChina, Shengli Oilfield Company, Zhongyuan Oilfield Company of Sinopec, for providing well data and permission to publish the results. Changrong Li is also appreciated for his kind help in the revision of the manuscript.

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

Table 1. Geological and geochemical characteristics of the main source rocks from the six representative petroliferous basins in China

| Basin location | Basin name | Basic features of representative basins | | | | Features of main source rocks | | | |
| | | Basin type | Basin area ($10^4$ km$^2$)/ maximum depth (m) | Heat flow (mW/m$^2$)/ geothermal gradient (°C/100 m) | National Ranking of reserves/ resources | Age and lithology | Organic matter abundance (TOC, %) | Organic matter type | Maximum measured maturity (Ro, %) |
|---|---|---|---|---|---|---|---|---|---|
| Western China | Tarim Basin | Complex superimposed basin | 53/ 9100 | 43.0/ 2.00 | 5/2 | Cambrian–Ordovician Carbonate | 0.2–5.0 | I–II | 3.7* |
| | Junggar Basin | Complex superimposed basin | 38/ 8900 | 45.0/ 2.30 | 4/5 | Permian Shale | 0.5–3.5 | I–II | 2.5 |
| Central China | Sichuan Basin | Superimposed basin | 26/ 7800 | 58.3/ 2.35 | 6/6 | Triassic Shale | 1.0–3.0 | II–III | 3.2 |
| | Ordos Basin | Superimposed basin | 37/ 6100 | 62.9/ 2.75 | 3/4 | Carboniferous–Permian Coal strata | 2.0–6.5 | II–III | 2.8 |
| Eastern China | Bohai Bay Basin | Fault Depression basin | 20/ 5800 | 64.8/ 3.20 | 1/1 | Paleogene Shale | 1.0–4.0 | I–II | 2.7 |
| | Songliao Basin | Rift-fault basin | 26/ 5400 | 69.0/ 4.00 | 2/3 | Jurassic–Cretaceous Shale | 1.0–4.0 | I–II | 3.6 |

* Ro =0.618*Ro$^B$ +0.40, Ro$^B$ is solid bitumen reflectance,%.

Table 2 Comparison of active source rock depth limits in the six petroliferous basins of China

| Research methods and related indicators for identifying ASDLs | | The maximum burial depth (D, m) and thermal maturity (Ro, %) corresponding to Active Source Rock Depth Limits | | | | | | |
|---|---|---|---|---|---|---|---|---|
| | | Tarim Basin | Junggar Basin | Sichuan Basin | Ordos Basin | Bohai bay Basin | Songliao Basin | The average values for six basins |
| The variation of element composition | H/C | 8970/3.5 | 8350/3.2 | – | – | 5800/3.5 | 5280/3.6 | 7100/3.4 |
| | O/C | 9050/3.6 | 8450/3.2 | – | – | 5740/3.4 | 5280/3.6 | 7130/3.4 |
| The variation of residual hydrocarbon | "A"/TOC | 9050/3.6 | 7850/3.0 | 7540/3.6 | 6450/3.3 | 5560/3.1 | 5330/3.7 | 6963/3.4 |
| | "$S_1$"/TOC | 9290/3.8 | 7960/3.0 | 7780/3.8 | 6500/3.4 | 5490/3.2 | 5400/3.9 | 7070/3.5 |
| The variation of hydrocarbon generation and expulsion | "$S_1+S_2$"/TOC | 9300/3.8 | 8200/3.0 | 7700/3.8 | 6600/3.4 | 5900/3.3 | 5400/3.9 | 7183/3.5 |
| | Ve | 9210/3.8 | 8200/3.0 | 7660/3.7 | 6520/3.4 | 5700/3.3 | 5500/4.0 | 7115/3.5 |
| The average values obtained from different methods in each basin | | 9145/3.7 | 8168/3.1 | 7670/3.7 | 6518/3.4 | 5698/3.3 | 5348/3.8 | 7094/3.5 |
| The data used for identifying ASDLs (sample number/well number) | | 2063/79 | 5353/351 | 460/27 | 1329/149 | 1193/69 | 3236/611 | Total: 13634/1286 |

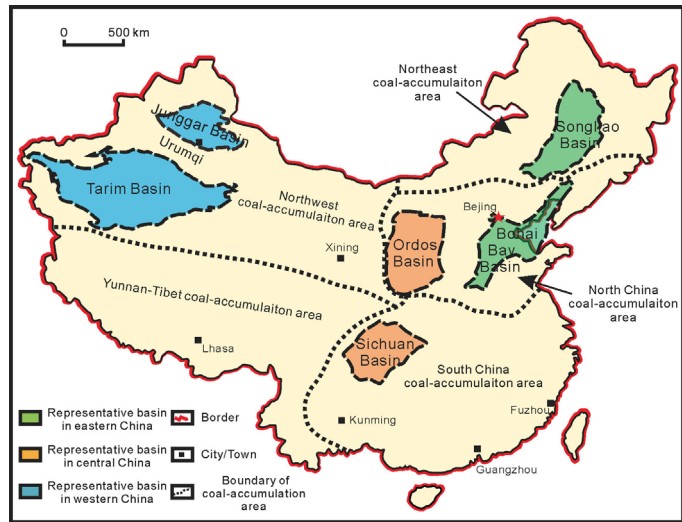

**Figure 1:** Location of the six representative petroliferous basins and five coal-accumulation areas in China. The studied petroliferous basins, plotted on the China mainland, are pigmented with different colors according to their locations in China. The five coal-accumulation areas, bounded by large geological structural belts, are mapped according to Zhu (2011).

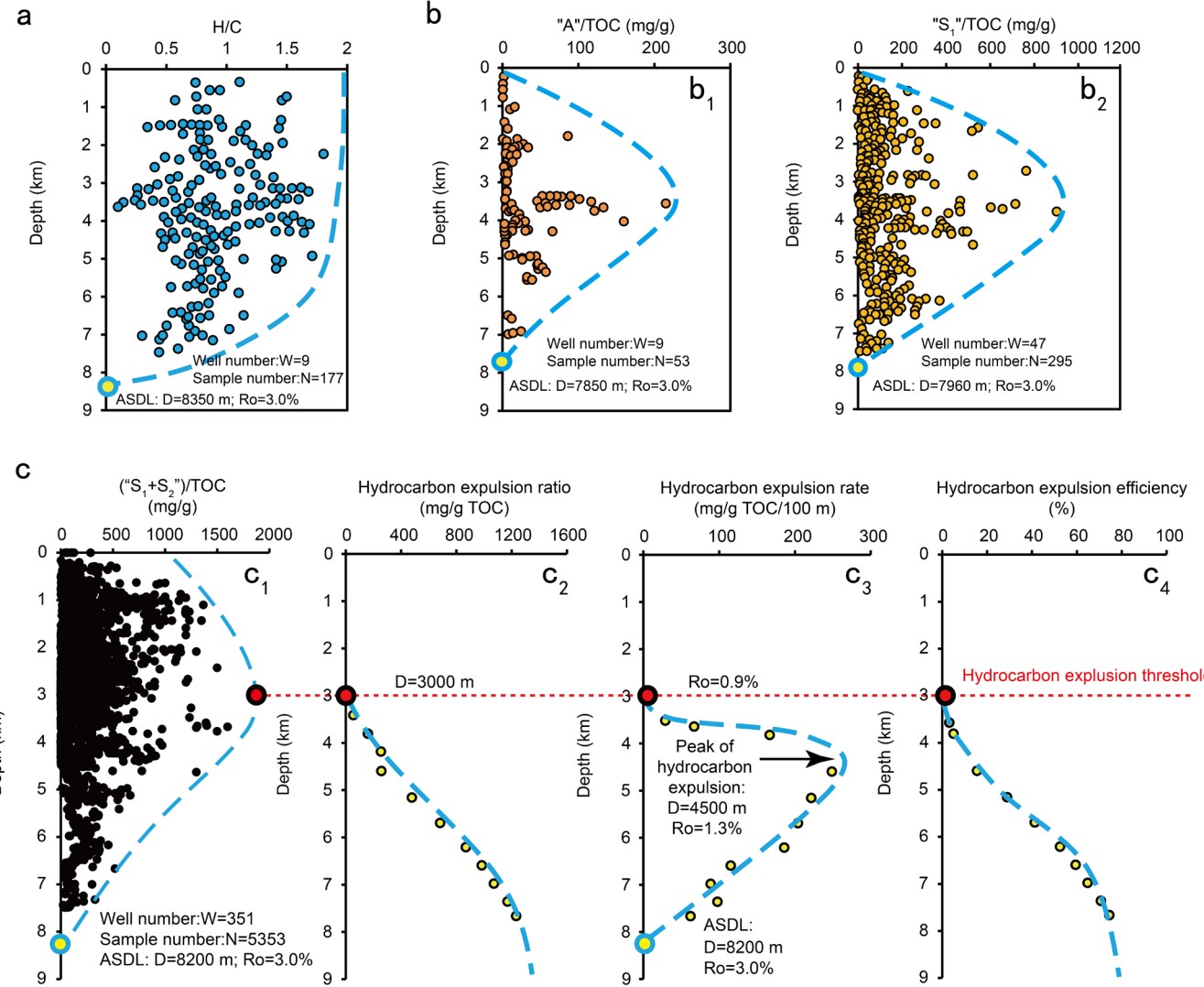

**Figure 2:** Identification of ASDL in the Junggar Basin using different indicators, including the variation of H/C ratios (a), residual hydrocarbon amounts (b), "$S_1 + S_2$"/TOC (c1), Qe (c2), Ve (c3) and Ke (c4) with depth.

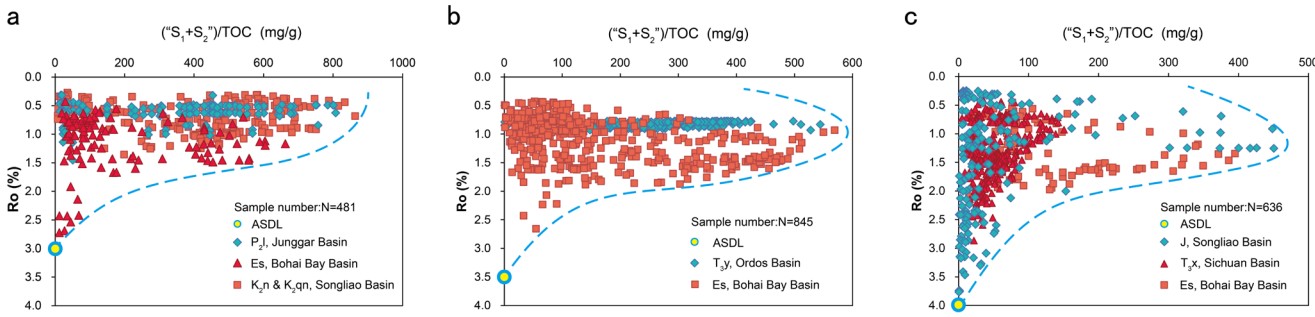

**Figure 3:** Effects of kerogen types on ASDLs represented by thermal maturity (Ro). From left to right are three plots of hydrocarbon generation potential index versus Ro for source rocks of Type I (a), Type II (b), and Type III (c).

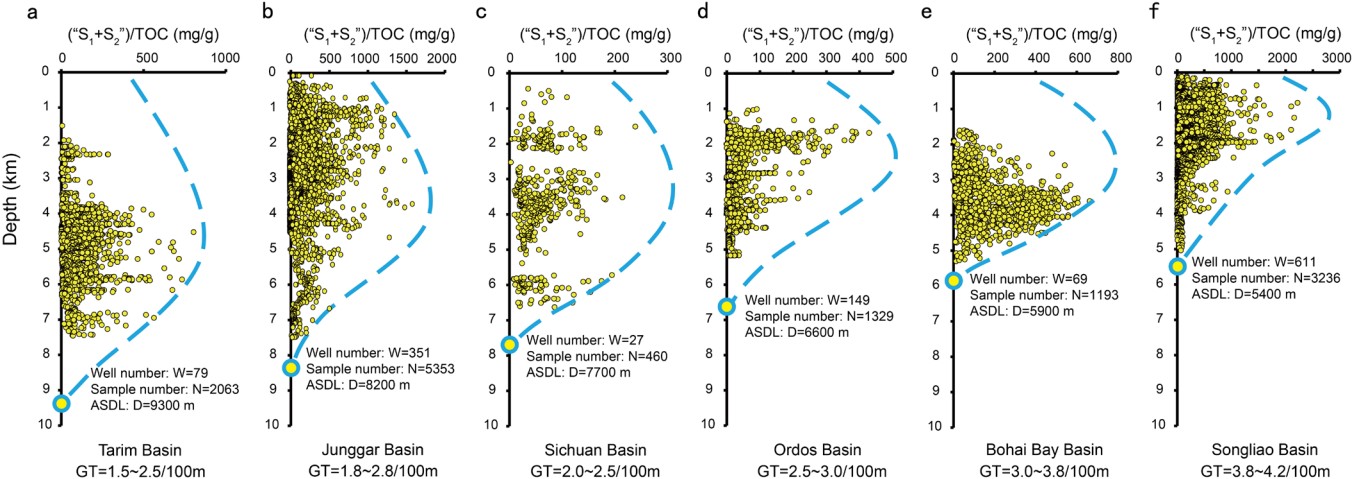

**Figure 4:** Variation of ASDLs in the six representative basins due to different heat flows. The ASDLs of different petroliferous basins are characterized by hydrocarbon generation potential index (represented by "$S_1 + S_2$"/TOC). From left to right, the heat flow (geothermal gradients) of each basin gradually increases, while the corresponding ASDL becomes shallower. a, Tarim Basin. b, Junggar Basin. c, Sichuan Basin. d, Ordos Basin. e, Bohai Bay Basin. f, Songliao Basin.

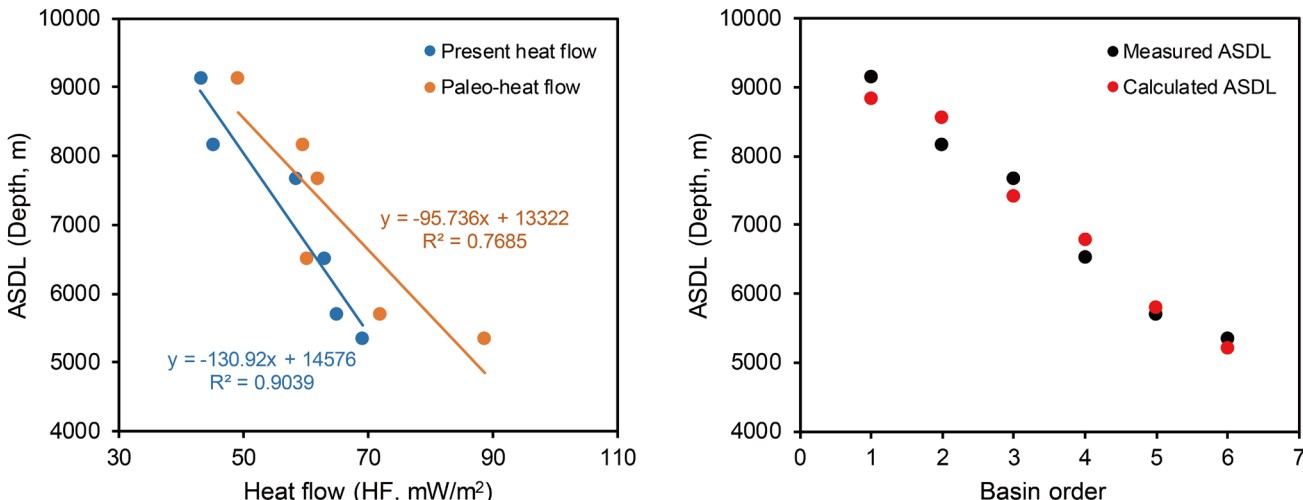

**Figure 5:** The quantitative relationships among the ASDL, heat flow and kerogen type for the six basins. a, relationship between ASDLs and heat flows. b, the comparison of the modelled depths through Eq. (1) and measured depths of the ASDLs. Basin order: 1. Tarim Basin; 2. Junggar Basin; 3. Sichuan Basin; 4. Ordos Basin; 5. Bohai Bay Basin; 6. Songliao Basin.

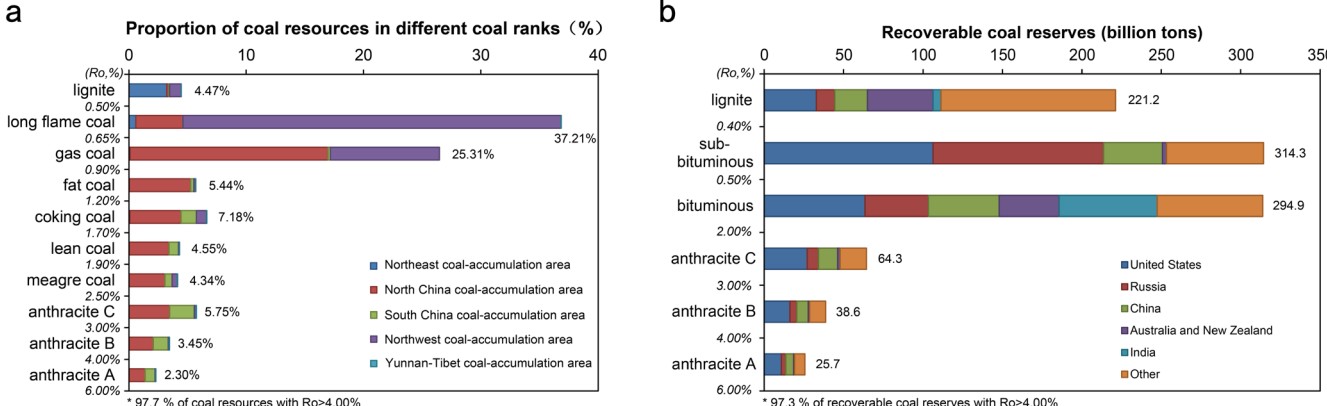

**Figure 6:** The variation of proved coal reserves with coal ranks in China and in the world. a, the proportion of proved coal reserves with different coal ranks in China (Data from CCRR, 1996; CNACG, 2016). The coal ranks are classified according to the Chinese standard, and the coal accumulation area is shown in Fig. 1. b, the recoverable coal reserves with different coal ranks around the word (Data from Conti et al., 2016). The coal ranks are classified according to international standard. The proved coal reserves of anthracite C, B and A are projected according to their variation trends.

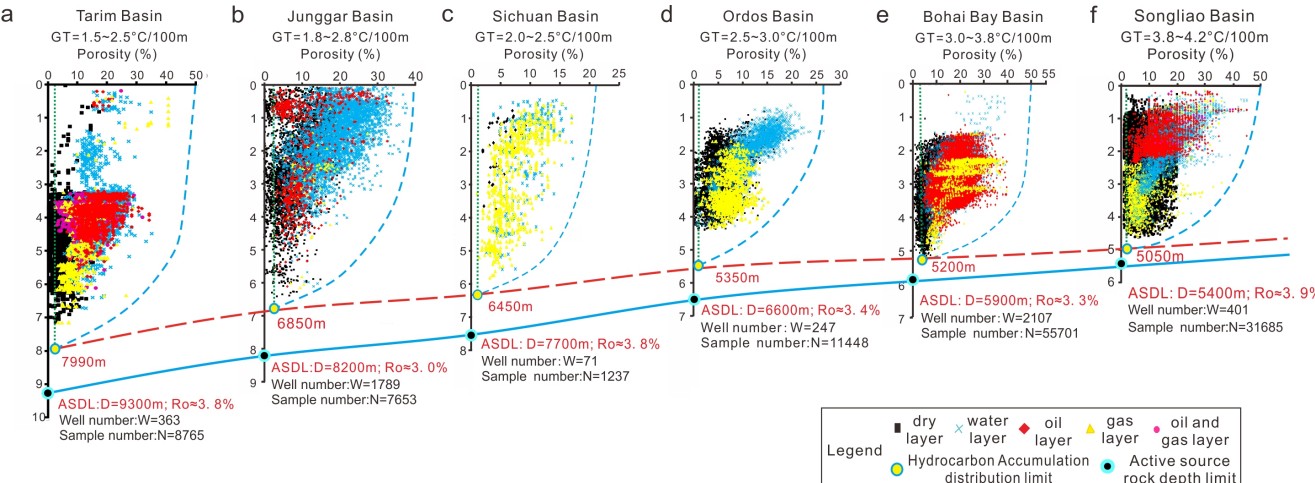

**Figure 7:** Hydrocarbon drilling results in the six representative petroliferous basins of China to show their relationships with the ASDLs and the HADLs. The results include 116,489 samples of target layers from 4,978 exploration wells in China. The blue dashed line represents the evolution of porosity with depth. Its intercept with the line of 2% porosity marks the HADL. The ASDL of each basin shown in this figure is represented by the value obtained from hydrocarbon generation potential index ("$S_1 + S_2$"/TOC) of each basin. From left to right: a, Tarim Basin. b, Junggar Basin. c, Sichuan Basin. d, Ordos Basin. e, Bohai Bay Basin. f, Songliao Basin. It is clear that the HADLs are always above the ASDLs.

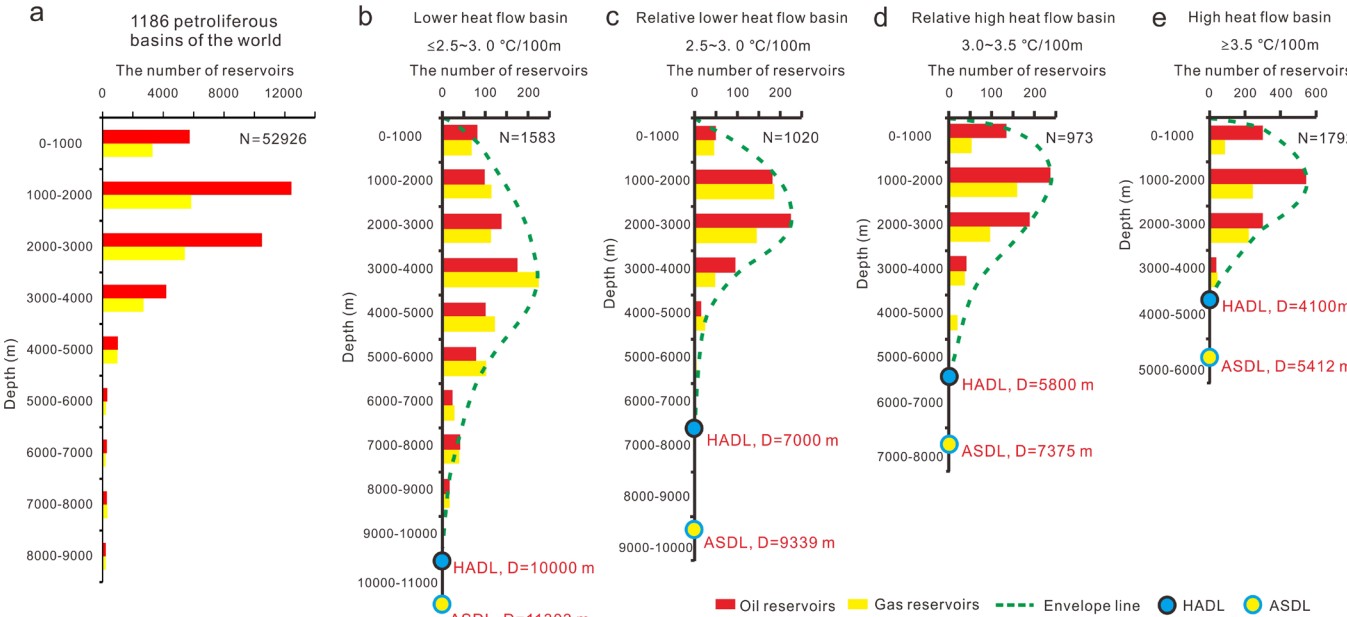

**Figure 8:** The vertical distribution of numbers of discovered hydrocarbon reservoirs and their relationships with ASDLs and HADLs in the worldwide 1,186 petroliferous basins. a, summation of proven reservoirs in the 1,186 basins. b, low heat flow basins (<25 mW/m²). c, relative low heat flow basins (25–40 mW/ m²). d, relative high heat flow basins (40–55 mW/ m²). e, high heat flow basins (55–70 mW/ m²). The intercept of the green dashed line on the vertical axis marks the HADL. The ASDL, shown in this figure, of each kind of basin with different heat flow is predicted by using the equation shown in Fig. 5.

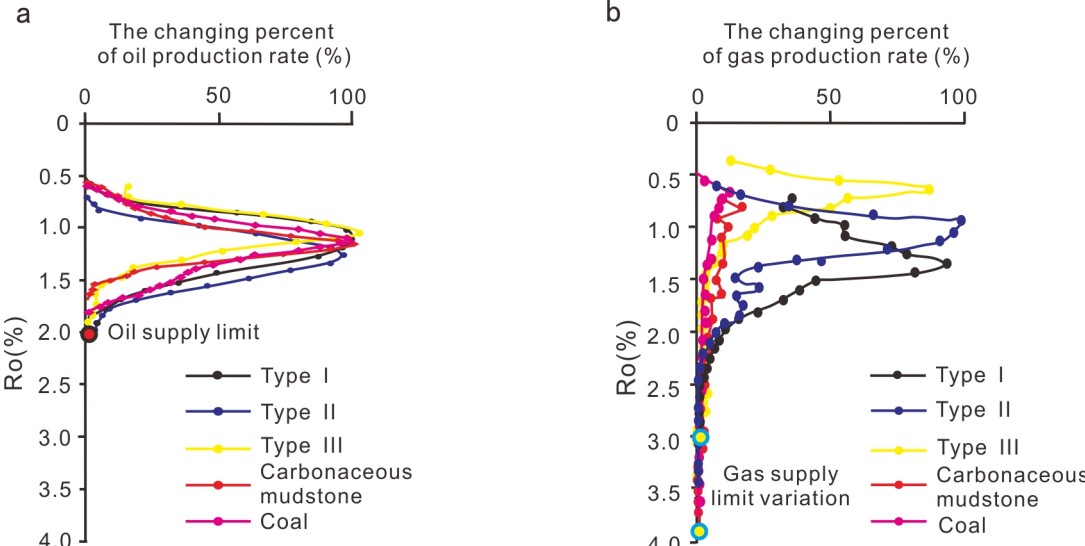

**Figure 9:** Investigation of ASDLs for different hydrocarbon types by high-temperature and high-pressure pyrolysis simulation. a, the variation of oil production rate with Ro and identification of ASDLo. b, the variation of gas production rate with Ro and identification of ASDLg.

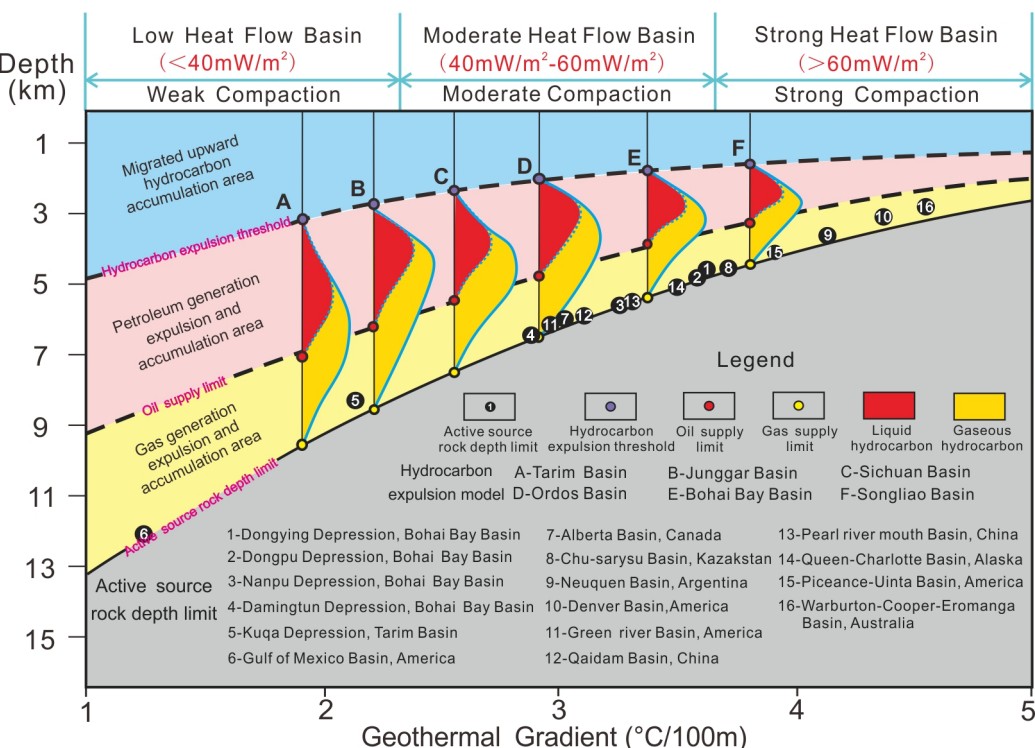

**Figure 10:** The pattern of ASDLs on the formation and distribution of hydrocarbon reservoirs in petroliferous basins. The upper blue area is favourable for the formation and distribution of conventional oil and gas resources, and hydrocarbons come from the underlying source rocks. The middle pink area is favourable for oil and gas generation, migration, and accumulation from source rocks in this area, mainly form conventional oil/gas reservoirs. The lower yellow area is favourable for natural gas generation, migration, and accumulation from source rocks, mainly form tight unconventional gas reservoirs.