# Peer review of "Active Source Rock Depth Limit and its Controlling on the Formation and Occurrence of Fossil Fuel Resources"

_Earth System Science Data, 2019_

## Referee Comment (RC1) · György Pogácsás (Referee) · 14 Aug 2019

General comments

The overall structure of the Xiongqi Pang et al "The Depth Limit for the Formation and Occurrence of Fossil Fuel Resources" article (https://www.earth-syst-sci-data-discuss.net/essd-2019-72/essd-2019-72.pdf) is fairly well structured and clear. The length of the paper is appropriate. The figures and tables are correct and good quality. The article is implemented by the Supplement (https://www.earth-syst-sci-data-discuss.net/essd-2019-72/essd-2019-72-supplement.pdf) and by the Data sets (https://doi.pangaea.de/10.1594/PANGAEA.900865).

Specific comments

[Figure]

The colour diagrams on both the figures of the article and on the figures of the Supplement are really beautiful and very informative. The Supplement figures (Fig. S1-Fig. S5) illustrate the identification of the Active Source Rock Depth Limit (ASDL) within the Tarim Basin, Sichuan Basin, Ordos Basin, Bohai Bay Basin, and Songliao Basin using different indicators. Supplement Fig. S6. illustrates the distribution of proven hydrocarbon reserves versus depth (and their relationships with ASDLs and HRDLs) within the Tarim Basin, Junggar Basin, Sichuan Basin, Ordos Basin, Bohai Bay Basin, and Songliao Basin.

The article and the data sets are more or less consistent. The excel spreadsheets of the Data sets at https://doi.pangaea.de/10.1594/PANGAEA.900865 (Fig.2.xlsx-Fig.9.xlsx) are strictly connected to the figures of the article as follows: Fig.2.xlsx to Fig.2; Fig.3.xlsx to Fig.3; Fig.4.xlsx to Fig.4; Fig.5.xlsx to Fig.5; Fig.6.xlsx to Fig.6; Fig.7.xlsx to Fig.7; Fig.8.xlsx to Fig.8; Fig.9.xlsx to Fig.9; whereas Fig.10 is connected both to Fig.8.xlsx and to the tables of the paper (Table 1. and Table 2). The Supplement figures referring to the Tarim Basin, Sichuan Basin, Ordos Basin, Bohai Bay Basin, and Songliao Basin, unfortunately are not corroborated by excel spreadsheets containing the data sets which were used for the construction of Fig. S1.-Fig. S5. Only in the case of the Junggar Basin figure (Fig.2) are available the excel spreadsheets (Fig.2.xlsx) at (https://doi.pangaea.de/10.1594/PANGAEA.900865) containing the geochemical data which were used to construct the diagrams on Fig.2.

The related Data sets at https://doi.pangaea.de/10.1594/PANGAEA.900865 contain an enormous number of geochemical data but the structure of the data sets excel spreadsheets (Fig.xlsx) does not seems to be the luckiest one. Although the different geochemical parameters of a source rock samples are strongly connected, the geochemical data cube of the article is separated and segmented: H/C versus depth, S1/TOC versus depth, A/TOC versus depth, (S1+S2)/TOC versus Ro, (S1+S2)/TOC versus depth etc. It does not help very much to understand the petroleum generation in the six representative Chinese basins. To provide additional references concerning the

petroleum systems of the Tarim Basin, Junggar Basin, Sichuan Basin, Ordos Basin, Bohai Bay Basin, Songliao Basin is strongly recommended. The article is focusing on questions related to depth limit of petroleum generation and occurrence. Geochemical data are definitely needed to answer these kind of questions and the paper itself seems to be appropriate to support the publication of the enclosed enormous geochemical data sets. Although question may arise concerning the need to use additional type (geology, geophysics etc.) of data as well to give even more precise answer concerning depth limit of petroleum generation within sedimentary basins. According to the article the Active Source Rock Depth Limit (ASDL) was studied basically by two more or less independent methods. The first one was a graphic method, constructing diagrams (H/C versus depth; A/TOC versus depth; S1/TOC versus depth; (S1 + S2)/TOC etc.) and "drawing dashed curve to envelope all the sample values and assuming the intercept of the dashed line on the vertical axis marks the ASDL". The second method applied linear equation as follows: ASDL = 16202 − 2.63 x HI − 139.46 x HF; "where, ASDL is the active source rock depth limit; HI is the hydrogen index value of the major source rock in a basin; HF is the average heat flow value of the given basin". In the case of the graphic method considerable uncertainty is related to the lack of ultra-deep wells why the drilled boreholes were terminated by some kilometres above the ASDL. In the case of the ASDL = 16202 − 2.63 x HI − 139.46 x HF equation the main uncertainty is related to the facts that heat flow values of the sedimentary basins are geologic time dependent and the thermal maturation is not the only function of HI and HF value, but it depends on other factors (time etc.) as well. Oil companies and research institutes all over the world apply sophisticated methods for modelling the source rock maturation versus time and versus depth, besides geochemical data using subsidence history reconstructions, deposition history models, compaction history models, thermal history calculations and so on. It is assumed comparison of the graphic method based and the linear equation (ASDL = 16202 − 2.63 x HI − 139.46 x HF) based results of the article with results of source rock maturation versus depth and time modelling would provide even more precise expectations concerning the depth limit of the ASDL within

the studied basins.

Technical comments Careful English language revision of the article by a native English college is strongly recommended.
* * *

---

## Short Comment (SC1) · 9 Sep 2019

I believe this is a very nice study. This manuscript proposed a new model and workflow to quantitative predict the depth limit of the fossil fuel resources in sedimentary basins, which has a wide implication in nowadays petroleum industry. Abundant oil exploration and production data from PetroChina and Sinopec were integrated in this study to illustrate their model. But I still have some comments and suggestions here: 1. Does the laboratory workflows for all these geochemical data as shown in figures 2, 3, and 4 are the same? Is there any deviation for different these data caused by different laboratory workflows? 2. How does the dashed envelope line be established in figures 2, 3, and 4? It's formed by handwriting or any professional software? It seems that some of these dashed envelope lines are not that well fit with the geochemical data.

3. How to preclude these outlier geochemical data? Is it possible that some outlier geochemical data (such as fig. 2b) are caused by detection deviation? As a whole, I congratulate the authors on a well-executed study.

---

## Short Comment (SC2) · 26 Sep 2019

As the conventional reservoirs have been mostly discovered and produced, the exploration work is gradually shifting to deeper, more changeling targets. This paper is aiming to provide possible vertical depth limits of potentially commercial reservoirs in a basin, and therefore lower the exploration risks. This study utilizes a large quantity of geochemistry and reservoir property data collected from 6 major, oil- and gas-producing basins in China as well as other major petroliferous basins around the world. The idea is the hydrocarbon generation and expulsion history of source rocks play a crucial role in the distribution of oil and gas reservoirs in a sedimentary basin. A variety of source rocks, including the Cambrian – Ordovician carbonate, Permian to Triassic marine shales, Carboniferous – Permian coals, and Paleogene lacustrine shales

are included in the study and these source rocks have remarkably different TOCs and thermal maturities. It's quite clear that this paper shows an objective of significant importance to the oil industry and lays a robust foundation in terms of dataset. This paper uses the hydrocarbon generation potential index ("S1+S2"/TOC) as the key method to determine the ASDLs. This method was proposed more than 10 years ago and has been successfully applied in a number of petroliferous basins in China (including those 6 basins in this study). From numerous previous studies, the changes of the hydrocarbon generation index with thermal maturity (Ro) or depth can be used to simulate the amount of hydrocarbon generated and expelled from source rocks in the geologic history. In the meantime, the maximum depth or thermal maturity can also be determined when the hydrocarbon potential approaches to zero. The results from this method are very consistent with those ASDLs determined from other geochemical parameters as well as the exploration results. In my opinion, there are two things need be cleared. First, how to determine the trends of the envelope lines indicating the change of the hydrocarbon potential index with depth or thermal maturity. Numerous studies have been done and published using this method in a number of basins in China. Detailed work has been conducted on source rocks with different kerogen types. The hydrocarbon generation potential indexes of these different types of source rocks all displayed similar trends (kind of like bell-shape) with depth or thermal maturity. But different source rocks showed very different hydrocarbon expulsion thresholds and efficiencies. These varying scenarios have been built on actual geochemical data and subsequently modeled by computer. Therefore, the envelop lines in this study are not randomly drawn but are guided by those well-established models. This has to been clearly explained because it is critical for the determination of the ASDLs. Second, this study proposes a linear equation to predict ASDL using basin heat flow data. It looks pretty good with the data from the 6 basins in the study, but there are only 6 data points. Statistically, the prediction is not be that robust. Two other things may also cause inaccurate predictions using this equation. First, basin heat flow (HF) can vary significantly over the geologic history. This paper did not clearly state the heat flow is the paleo-HF or

present HF of the basins. Second, I assume the depth is the present-day burial depth, which can be very different from the maximum burial depth in the basin history, especially foreland basins (e.g. Appalachian Basin, USA). The late-stage uplifting in the Appalachian Basin can be up to thousands of meters. Marcellus Shale in the northeast Pennsylvania, for example, shows very high thermal maturity (3 – 4.5% Ro) and in some area, has passed the ASDL, but the present-day depth generally is less than 3 – 4 km. In contrast, some rifted basins have never undergone any uplifting at all. Given the complex geologic histories of some basins, thermal maturity (Ro) appears to be more reliable (also pointed out in this paper). Although different types of basins are included in the study and the predicted ASDLs appear to be very reasonable, it would be nice to run some blind test including some data from basins in other continents. Also, it would be great to add some assumptions regarding the possible limitations of this application. In summary, this paper has a well-defined objective, a huge amount of data, a well proved methodology, and very solid results. The publication of this paper will bring a remarkable impact to the oil industry and the new findings from this paper surely will help lower exploration risks.

---

## Short Comment (SC3) · 29 Sep 2019

Fossil fuel resources play an important role in the modern life. As the increasing demand of hydrocarbon, how to streamline the exploration is of great significance. As is well known, the vertical distribution of hydrocarbon varies from several hundred meters to tens of kilometers, which can bring high risks to the oil and gas assessment and exploration. This paper introduces an innovative concept: Active Source Rock Depth Limits (ASDL), which defines the maximum burial depth for source rocks to generate hydrocarbon in sedimentary basins. The concept can significantly provide guidance for hydrocarbon assessment and exploration. Detailed examples from six major petroliferous basins in China have been presented. The method used to define ASDL is very solid, parameters applied in this paper like H/C ratios, "A"/TOC, "S1+S2"/TOC are

widely used in organic geochemistry. The result shows the existence of ASDL, and it is comparable across these basins regarding thermal maturity. Apart from the basins in China, the paper also shows the concept of ASDL is applicable to the basins all over the world, which strongly support the authors' idea. Moreover, the controlling factors of ASDL has been investigated in the paper and the authors point out the ASDL is mainly constrained by the heat flow of the basin and types of organic matter. Quantitative research in hydrocarbon geology has always been a challenging task, however, through geochemical and mathematical analysis, the quantitative relations between the above two controlling factors and ASDL is established in this paper, which is a good try. At the ending part of the paper, the authors come up with the idea that the petroliferous basin can be vertically divided into three parts based on HET, oil and gas supply limits, and each part has its own types of hydrocarbon reservoirs. The idea provide a more efficient way to target hydrocarbon reservoirs, which is very interesting and hopefully we can see more details about this in the future.

Honestly speaking, I believe the work of this paper is very innovative and the result is very solid, and it provides us with a new perspective when doing fossil fuel exploration, which definitely deserves to be published. However, I still have few suggestions and comments for this paper: 1.Geochemical data and hydrocarbon reservoir parameters from six representative petroliferous basins in China are quite enough. I am not sure whether the corresponding dataset for basins from worldwide is accessible, like Persian Gulf Basin, West Canada Basin, North Sea Basin. If one or few detailed examples of above mentioned basins can be presented, it would be the icing on the cake. 2.In the part of quantitative prediction of ASDL, the predicted ASDL can be expressed by HI and HF, where HI is the proxy of organic matter types. As we all know, there are some other parameters can represent kerogen type, why the HI is selected to established the quantitative relation in this paper? 3.In part 3.1, the average thermal maturity level is regarded as the identification criterion for ASDL in general geological settings. However, in part 3.3, the quantitative prediction of ASDL is express by active source rock depth limit. As is known, the thermal evolution of organic matter is a chemical

process and it is a function of time and heat. Therefore, I believe if we want express ASDL by depth, it is better to take the age of source rock into consideration when doing quantitative investigation. 4.In part 3.3, it seems that the HF used here is the value of nowadays. But HF varies in geological history and can be influenced by tectonic events. For example, the current heat flow in Basin and Range is high due to the recent tectonic extension. Therefore, maybe an average heat flow from the time when the source rock deposited to present is a better. 5.In conclusion 1 and 2, what is meaning of ASRL? Is this a typo or Does it represent something else?

---

## Referee Comment (RC2) · Ludden John (Referee) · 30 Sep 2019

This manuscript is a useful contribution of data from numerous oil and gas wells in China and a world-wide compilation. It provides very useful data that is on-line in the PANGEA data-base. I would expect these data to be used by a number of users both from academia and industry and this is a good example of open data.

The only thing I struggle with is the static view of basins in these models. The oil and gas community model the source rocks as expelling gas and oil simply due to subsidence and compaction - as in Figure 9. This is largely because the oil and gas producers are mainly interested in the reservoir rather than the source - so they focus on the trapping process. Here the authors attempt to define limits for expulsion and

these vary with heat flow of the basin as one would expect. However, there are fluxes of fluids, water rich and saline, traversing these basins and the role of these in dissolving organic matter and redistributing it has largely been ignored. I refer the authors to a recent paper which I was associated with in which we show that Pb- in oil comes from older and clearly deeper sources and is a mixture of components Lead isotopes as tracers of crude oil migration within deep crustal fluid systems Earth and Planetary Science Letters, Volume 525, 1 November 2019, Article 115747

There is some minor grammatical structuring that the authors should ensure for clarity in particular in the section - last paragraph on characteristics of ASDL.

---

## Short Comment (SC4) · 30 Sep 2019

I find this manuscript is generally well written, structured, and illustrated. Some of the results are very interesting and thought-provoking. It puts forward the concept of "Active Source Rock Depth Limits (ASDL)", and try to characterize the vertical depth distribution of discovered reservoirs. A huge of data has been systematically compiled around the world, especially the six key basins in China. The use of four methods to characterize and corroborate the ASDL, including the possibility to be used around the world, makes it more convincing. The controls on the ASDL are also explored and a quantitative model was established to predict the results. The study is very meaningful in the way that it is the first systematic attempt to work on the relationship of depths and hydrocarbon reservoirs, especially on such a huge scale around the world, with

so much data. It brings this topic to our attention which should be studied before. The topic has great scientific values. It could help us to better understand why at a shallow depth there are no reservoirs in a basin whereas in some other basins, reservoirs are found in a much deeper layer. If proven, it will also help us to determine whether to drill a well to a certain depth.

During my reading of this MS, I had several questions or suggestions for the authors. If they could help with them, I would really appreciate it.

1. The ASDL was mostly established with the data of six basins in China, although data of basins around the world (IHS, 2010) were later used to verify the ASDL. Is it possible to incorporate some basins outside of China in the process of establishing the model? I totally understand this is totally a data issue and the authors may not be able to get enough systematic data of basin around the world as in China. However, this could be an improvement, if possible. 2. In section 3.3, the authors used the average values (heat value, depth, HI) of each basin to verify the model of ASDL, it is generally OK and understandable as there must be lots of data for each basin. However, is there still any possibility that some outliers may be present? If yes, how to explain them? 3. In section 3.4, the authors proposed the concept of Hydrocarbon Reservoir Depth Limit (HRDL), mentioning that "at some depth (Hydrocarbon Reservoir Depth Limit), the probability of drilling oil or gas reservoirs decreases to zero", and talks a bit about the relationship between HRDL and ASDL. I think this is an important part as for a hydrocarbon reservoir to form, it requires both hydrocarbons from source rocks and reservoirs rock to accumulate. Unfortunately, very little was discussed on the HRDL in this point. If possible, could more details be added on this?

Personally, I am very attracted to this study and would like to continue to catch up with the progress. I also appreciate a lot the hard work and innovativeness of this study that the authors have put into.

---

## Author Response (AR1)

Dear Editors and Reviewers,

We are deeply grateful for the editorial board's consideration of our manuscript, and we would also like to express our deep appreciation to reviewers for two referee comments and four short comments. Their constructive suggestions and insightful comments really help us improve the readability, quality, and rigour of the manuscript. We have revised the manuscript carefully following the suggestions. In this response, we have clarified what we have done to tackle the issues proposed by the reviewers. For clarity's sake, the comments are in **bold** type, and the responses including additional works that have been implemented into the manuscript are in normal type with blue colour.

***Referee Comment #1 by György Pogácsás***

**General Comments: The overall structure of the Xiongqi Pang et al "Active Source-rock Depth Limit and Its Controlling on the Formation and Occurrence of Fossil Fuel Resources" article is fairly well structured and clear. The length of the paper is appropriate. The figures and tables are correct and good quality. The article is implemented by the Supplement and by the Data sets. The colour diagrams on both the figures of the article and on the figures of the Supplement are really beautiful and very informative. The Supplement figures (Fig. S1-Fig. S5) illustrate the identification of the Active Source Rock Depth Limit (ASDL) within the Tarim Basin, Sichuan Basin, Ordos Basin, Bohai Bay Basin, and Songliao Basin using different indicators. Supplement Fig. S6. illustrates the distribution of proven hydrocarbon reserves versus depth (and their relationships with ASDLs and HADLs) within the Tarim Basin, Junggar Basin, Sichuan Basin, Ordos Basin, Bohai Bay Basin, and Songliao Basin.**

**(1) The article and the data sets are more or less consistent. The excel spreadsheets of the Datasets (Fig.2.xlsx-Fig.9.xlsx) are strictly connected to the figures of the article as follows: Fig.2.xlsx to Fig.2; Fig.3.xlsx to Fig.3; Fig.4.xlsx to Fig.4; Fig.5.xlsx to Fig.5; Fig.6.xlsx to Fig.6; Fig.7.xlsx to Fig.7; Fig.8.xlsx to Fig.8; Fig.9.xlsx to Fig.9; whereas Fig.10 is connected both to Fig.8.xlsx and to the tables of the paper (Table 1. and Table 2). The Supplement figures referring to the Tarim Basin, Sichuan Basin, Ordos Basin, Bohai Bay Basin, and Songliao Basin, unfortunately are not corroborated by excel spreadsheets containing the data sets which were used for the construction of Fig. S1.-Fig. S5. Only in the case of the Junggar Basin figure (Fig.2) are available the excel spreadsheets**

**(Fig.2.xlsx) at (https://doi.pangaea.de/10.1594/PANGAEA.900865) containing the geochemical data which were used to construct the diagrams on Fig.2.**

Response: Thanks for your suggestion. We have supplemented the geochemical datasets that can be utilized to construct the figures of Fig.S1-Fig.S5. It should be clarified that during the process of organizing the uploaded data, we found serious errors in previously supplemented figures and we are very sorry for that. In the subfigure (a) and (b) of Fig.S1-Fig.S5, the ordinate should be the thermal maturity, i.e. Ro, that the depth. We have corrected this in the newly supplemented figures, and have also uploaded the relevant data. Ro is chosen because it is closely related to the ASDL, and hydrocarbon generation and expulsion. Furthermore, according to the reviewers' suggestion, the original data was further proofread during the process of organizing the data, and the number of data points has changed slightly due to the screening of outliers. But the overall trends have not been changed.

Changes: We have supplemented the geochemical datasets that can utilized to construct the figures of Fig.S1-Fig.S5. The supplementary files are also updated during the revision. Please find the updated datasets and figures in the attached supplementary file.

**(2) The related Data sets at https://doi.pangaea.de/10.1594/PANGAEA.900865 contain an enormous number of geochemical data but the structure of the data sets excel spreadsheets (Fig.xlsx) does not seems to be the luckiest one. Although the different geochemical parameters of a source rock samples are strongly connected, the geochemical data cube of the article is separated and segmented: H/C versus depth, S1/TOC versus depth, A/TOC versus depth, (S1+S2)/TOC versus Ro, (S1+S2)/TOC versus depth etc. It does not help very much to understand the petroleum generation in the six representative Chinese basins.**

Response: The primary purpose for constructing such kind of structure is for the convenience of the scientific and industrial community to reproduce the figures in the manuscript quickly and to reutilize these data efficiently. The reviewer, however, did pinpoint an important issue we have ignored, and therefore thank you very much for your comment. For the data that have been uploaded to the PANGAEA, it may be difficult to change the structure of these data, but as for the new data, we have followed the reviewer's suggestion to put every basin's geochemical data (i.e. H/C versus depth,

S1/TOC versus depth, A/TOC versus depth, (S1+S2)/TOC versus Ro, (S1+S2)/TOC versus depth ) in a same excel sheet, so as to correlate the data with petroleum generation more closely.

Changes: We have followed the reviewer's suggestion to put every basin's geochemical data in a same excel sheet. Please find the datasets in the attached supplementary file.

**(3) To provide additional references concerning the petroleum systems of the Tarim Basin, Junggar Basin, Sichuan Basin, Ordos Basin, Bohai Bay Basin, Songliao Basin is strongly recommended. The article is focusing on questions related to depth limit of petroleum generation and occurrence. Geochemical data are definitely needed to answer these kinds of questions and the paper itself seems to be appropriate to support the publication of the enclosed enormous geochemical data sets. Although question may arise concerning the need to use additional type (geology, geophysics etc.) of data as well to give even more precise answer concerning depth limit of petroleum generation within sedimentary basins.**

Response: Thank you very much for your suggestion. The additional references, especially for geology and geophysics, with respect to petroleum systems of the six representative basins have been provided in the revised manuscript.

Changes: In the section 3.1, we have briefly introduced the petroleum system of the Junggar basin and provided the references regarding the petroleum systems of other five basins. The detailed revisions can be found at the end of this response, which is attached as a marked-up manuscript.

The references added are listed here:
1. Wang, S., He, L., & Wang, J.: Thermal regime and petroleum systems in Junggar Basin, northwest China. Physics of the Earth and Planetary Interiors, 126(3-4), 237-248, doi:10.1016/S0031-9201(01)00258-8, 2001.
2. Cao, J., Zhang, Y., Hu, W., Yao, S., Wang, X., Zhang, Y., & Tang, Y.: The Permian hybrid petroleum system in the northwest margin of the Junggar Basin, northwest China. Marine and Petroleum Geology, 22(3), 331-349, doi:10.1016/j.marpetgeo.2005.01.005, 2005.
3. Chen, Z., Zha, M., Liu, K., Zhang, Y., Yang, D., Tang, Y., Tang, Y., Wu, K., & Chen,

Y.: Origin and accumulation mechanisms of petroleum in the Carboniferous volcanic rocks of the Kebai Fault zone, Western Junggar Basin, China. Journal of Asian Earth Sciences, 127, 170-196, doi:10.1016/j.jseaes.2016.06.002, 2016.

4. Wang, Y., Yang, R., Song, M., Lenhardt, N., Wang, X., Zhang, X., Yang, S., Wang, J., & Cao, H.: Characteristics, controls and geological models of hydrocarbon accumulation in the Carboniferous volcanic reservoirs of the Chunfeng Oilfield, Junggar Basin, northwestern China. Marine and Petroleum Geology, 94, 65-79, doi:10.1016/j.marpetgeo.2018.04.001, 2018.

5. Zhou, Y., & Littke, R.: Numerical simulation of the thermal maturation, oil generation and migration in the Songliao Basin, Northeastern China. Marine and Petroleum Geology, 16(8), 771-792, doi:10.1016/S0264-8172(99)00043-4, 1999.

6. Xiao, X. M., Zhao, B. Q., Thu, Z. L., Song, Z. G., & Wilkins, R. W. T.: Upper Paleozoic petroleum system, Ordos Basin, China. Marine and Petroleum Geology, 22(8), 945-963, doi:10.1016/j.marpetgeo.2005.04.001, 2005.

7. Wu, S. X., Jin, Z. J., Tang, L. J., & Bai, Z. R.: Characteristics of Triassic petroleum systems in the Longmenshan foreland basin, Sichuan province, China. Acta Geologica Sinica‐English Edition, 82(3), 554-561, doi:10.1111/j.1755-6724.2008.tb00606.x, 2008.

8. Ping, H., Chen, H., & Jia, G.: Petroleum accumulation in the deeply buried reservoirs in the northern Dongying Depression, Bohai Bay Basin, China: New insights from fluid inclusions, natural gas geochemistry, and 1-D basin modeling. Marine and Petroleum Geology, 80, 70-93, doi:10.1016/j.marpetgeo.2016.11.023, 2017.

9. Zhu, G., Cao, Y., Yan, L., Yang, H., Sun, C., Zhang, Z., Li, T., & Chen, Y.: Potential and favorable areas of petroleum exploration of ultra-deep marine strata more than 8000 m deep in the Tarim Basin, Northwest China. Journal of Natural Gas Geoscience, 3(6), 321-337, doi:10.1016/j.jnggs.2018.12.002, 2018.

**(4) According to the article the Active Source Rock Depth Limit (ASDL) was studied basically by two more or less independent methods. The first one was a graphic method, constructing diagrams (H/C versus depth; A/TOC versus depth; S1/TOC versus depth; (S1 + S2)/TOC etc.) and "drawing dashed curve to envelope all the sample values and assuming the intercept of the dashed line on the vertical axis marks the ASDL". The second method applied linear equation as follows: ASDL = 16202 - 2.63 x HI + 139.46 x HF;  "where, ASDL is the active source rock depth limit; HI is the hydrogen index value of the major source rock in a basin; HF is the average heat flow value of the given basin". In the case of**

the graphic method considerable uncertainty is related to the lack of ultra-deep wells why the drilled boreholes were terminated by some kilometres above the ASDL. In the case of the ASDL = 16202 - 2.63 x HI + 139.46 x HF equation the main uncertainty is related to the facts that heat flow values of the sedimentary basins are geologic time dependent and the thermal maturation is not the only function of HI and HF value, but it depends on other factors (time etc.) as well. Oil companies and research institutes all over the world apply sophisticated methods for modelling the source rock maturation versus time and versus depth, besides geochemical data using subsidence history reconstructions, deposition history models, compaction history models, thermal history calculations and so on. It is assumed comparison of the graphic method based and the linear equation (ASDL = 16202 - 2.63 x HI + 139.46 x HF) based results of the article with results of source rock maturation versus depth and time modelling would provide even more precise expectations concerning the depth limit of the ASDL within the studied basins.

Response: Thank you very much for your suggestion. We totally agree with the reviewer that these uncertainties may play an important role in affecting the depth limit of the ASDL. Therefore, we have modified the manuscript from the following aspects. For the first case that uncertainties may be caused by the lack of datasets from ultra-deep wells, the ASDL can be identified by extrapolation according to the variation regularity of these data with depth, and such kind of regularity is derived from the actual data from different basins. For the second case that uncertainties may be brought in by the quantitative equation, we have re-written the section 3.3 in the revised manuscript that the proposed equation is only used to elucidate that there exists a relationship between the ASDL and the heat flow and organic matter type. This is because, as the reviewer suggested, the influence factors of the ASDL are not fully incorporated into the equation. We have already mentioned in the manuscript that the ASDL is not only influenced by the heat flow and organic matter type, but also influenced by the stratigraphic age and tectonic uplift. However, to set up the equation with four independent variables by using our database with only 6 basins is difficult and impossible. Therefore, we have supposed in the revised manuscript that the proposed equation cannot utilized to predict precisely the ASDL of source rocks in other basins except the six representative basins of China, especially for basins out of China. On the other hand, basin modelling and some other integrated analysis methods, just as the reviewer suggested, should be applied if readers want to decide the depth limit of ASDL which should be correspondence to the criteria provided by our

study (i.e. Ro=3.5% ± 0.5%).

Changes: We have added relevant content in section 3.1 and re-written section 3.3 according to the above response, and the detailed revisions can be found at the end of this response, which is attached as a marked-up manuscript.

**(5) Careful English language revision of the article by a native English college is strongly recommended.**

Response: Thanks for your suggestion. We have modified the manuscript with the help of a native English researcher.

Changes: The manuscript has been polished by a native English researcher, and please find the marked-up manuscript at the end of this response to see the detailed revisions.

*Referee Comment #2 by Ludden John*

**General Comments: This manuscript is a useful contribution of data from numerous oil and gas wells in China and a world-wide compilation. It provides very useful data that is on-line in the PANGEA data-base. I would expect these data to be used by a number of users both from academia and industry and this is a good example of open data.**

**(1) The only thing I struggle with is the static view of basins in these models. The oil and gas community model the source rocks as expelling gas and oil simply due to subsidence and compaction - as in Figure 9. This is largely because the oil and gas producers are mainly interested in the reservoir rather than the source - so they focus on the trapping process. Here the authors attempt to define limits for expulsion and these vary with heat flow of the basin as one would expect. However, there are fluxes of fluids, water rich and saline, traversing these basins and the role of these in dissolving organic matter and redistributing it has largely been ignored. I refer the authors to a recent paper which I was associated with in which we show that Pb- in oil comes from older and clearly deeper sources and is a mixture of components Lead isotopes as tracers of crude oil migration within deep crustal fluid systems Earth and Planetary Science Letters, Volume 525, 1 November 2019, Article 115747.**

Response: Thanks for your suggestion. It is true that source rocks can expel hydrocarbons as the consequence of thermal maturation with increasing depth, and there are a variety of software to model such kind of process. It is hard to say, however, that the oil and gas companies are mainly interested in the reservoirs rather than the source rocks. Conversely, these companies are paying more attention to the source rocks than ever before, which may be induced by the U.S. shale revolution to a large extent. On the other hand, we agree with the reviewer that the fluid fluxes have influences on the generation, preservation and redistribution of organic matter in source rock, and then have influences on its ASDL. Therefore, we have discussed the relevant problems and added relative references in the revised manuscript.

Changes: According to the reviewer's suggestion, we have added created a new section 3.2.3 to discuss the influence of deep thermal fluids and overpressure on the ASDL. The detailed revisions can be found at the end of this response, which is attached as a marked-up manuscript.

The references added are listed here:

1. McTavish, R. A.: The role of overpressure in the retardation of organic matter maturation. Journal of Petroleum Geology, 21(2), 153-186, doi:10.1111/j.1747-5457.1998.tb00652.x, 1998.

2. Hao, F., Zou, H., Gong, Z., Yang, S., & Zeng, Z.: Hierarchies of overpressure retardation of organic matter maturation: Case studies from petroleum basins in China. AAPG bulletin, 91(10), 1467-1498, doi:10.1306/05210705161, 2007.

3. Fetter, N., Blichert-Toft, J., Ludden, J., Lepland, A., Borque, J. S., Greenhalgh, E., Garcia, B., Edwards, D., Télouk, P., & Albarède, F. (2019). Lead isotopes as tracers of crude oil migration within deep crustal fluid systems. Earth and Planetary Science Letters, 525, 115747.

4. Rullkötter, J., Leythaeuser, D., Horsfield, B., Littke, R., Mann, U., Müller, P. J., Schaefer, R.G., Schenk, H.-J., Schwochau, K., & Witte, E. G.: Organic matter maturation under the influence of a deep intrusive heat source: a natural experiment for quantitation of hydrocarbon generation and expulsion from a petroleum source rock (Toarcian shale, northern Germany). Organic Geochemistry, 13(4-6), 847-856, doi:10.1016/0146-6380(88)90237-9, 1988.

5. Zhu, D., Liu, Q., Jin, Z., Meng, Q., & Hu, W.: Effects of deep fluids on hydrocarbon generation and accumulation in Chinese petroliferous basins. Acta Geologica Sinica‐English Edition, 91(1), 301-319, doi:10.1111/1755-6724.13079, 2017.

**(2) There is some minor grammatical structuring that the authors should ensure for clarity in particular in the section - last paragraph on characteristics of ASDL.**

Response: Thank you for the suggestion on language improving. We have modified the manuscript with the help of a native English researcher.

Changes: The manuscript has been polished by a native English researcher, and please find the marked-up manuscript at the end of this response to see the detailed revisions.

*Short Comment #1 by Shuang Xiao*
**General Comments: I believe this is a very nice study. This manuscript proposed a new model and workflow to quantitative predict the depth limit of the fossil fuel resources in sedimentary basins, which has a wide implication in nowadays petroleum industry. Abundant oil exploration and production data from PetroChina and Sinopec were integrated in this study to illustrate their model. But I still have some comments and suggestions here:**

**(1) Does the laboratory workflows for all these geochemical data as shown in figures 2, 3, and 4 are the same? Is there any deviation for different these data caused by different laboratory workflows?**

Response: Thanks for your comment. All these geochemical data shown in Figures 2, 3, 4 are obtained following the same workflows which are in accordance with Chinese Petroleum and Natural Gas Industry Standard of P.R. China. We admit that there may be deviations due to the experimental errors, but the deviations of these data are within the controllable range since the workflows strictly follow a suit of standards.

Changes: According to the above response to the reviewer, there is no change in the revised manuscript related to this comment.

**(2) How does the dashed envelope line be established in figures 2, 3, and 4? It's formed by handwriting or any professional software? It seems that some of these dashed envelope lines are not that well fit with the geochemical data.**

Response: Thanks for your comment. As described in the manuscript, the envelope

line is drawn by including all sample values. Instead of randomly drawing, the envelope line drawn by handwriting on the model of numerous previous studies investigating the changing trend of hydrocarbon generation potential index and actual geological data.

Changes: We have added a part of content in section 3.1 clarifying the determination of envelope lines, and the detailed revisions can be found at the end of this response, which is attached as a marked-up manuscript.

**(3) How to preclude these outlier geochemical data? Is it possible that some outlier geochemical data (such as fig. 2b) are caused by detection deviation?**

Response: Thanks for your comment. The outlier is a data that differs significantly from other observations, and according to this trait, some outlier data have already been discarded during the compilation of these geochemical data. However, in Fig. 2b1, for example, the highest value is not significantly different from the others. On the other hand, even if we exclude those "outlier" data as supposed by Shuang Xiao, a clear trend still stands with remaining data and the turning point corresponding to hydrocarbon expulsion threshold does not change. Therefore, we argue that these data are not outliers.

Changes: According to the above response to the reviewer, there is no change in the revised manuscript related to this comment.

*Short Comment #2 by Jie Zhou*
**General Comments: As the conventional reservoirs have been mostly discovered and produced, the exploration work is gradually shifting to deeper, more changeling targets. This paper is aiming to provide possible vertical depth limits of potentially commercial reservoirs in a basin, and therefore lower the exploration risks. This study utilizes a large quantity of geochemistry and reservoir property data collected from 6 major, oil- and gas-producing basins in China as well as other major petroliferous basins around the world. The idea is the hydrocarbon generation and expulsion history of source rocks play a crucial role in the distribution of oil and gas reservoirs in a sedimentary basin. A variety of source rocks, including the Cambrian – Ordovician carbonate, Permian to Triassic marine shales, Carboniferous – Permian coals, and Paleogene lacustrine shales are included in the study and these source rocks have remarkably different TOCs and thermal maturities. It's quite clear that this paper**

shows an objective of significant importance to the oil industry and lays a robust foundation in terms of dataset. This paper uses the hydrocarbon generation potential index ("S1+S2"/TOC) as the key method to determine the ASDLs. This method was proposed more than 10 years ago and has been successfully applied in a number of petroliferous basins in China (including those 6 basins in this study). From numerous previous studies, the changes of the hydrocarbon generation index with thermal maturity (Ro) or depth can be used to simulate the amount of hydrocarbon generated and expelled from source rocks in the geologic history. In the meantime, the maximum depth or thermal maturity can also be determined when the hydrocarbon potential approaches to zero. The results from this method are very consistent with those ASDLs determined from other geochemical parameters as well as the exploration results. In my opinion, there are two things need be cleared:

(1) First, how to determine the trends of the envelope lines indicating the change of the hydrocarbon potential index with depth or thermal maturity. Numerous studies have been done and published using this method in a number of basins in China. Detailed work has been conducted on source rocks with different kerogen types. The hydrocarbon generation potential indexes of these different types of source rocks all displayed similar trends (kind of like bell-shape) with depth or thermal maturity. But different source rocks showed very different hydrocarbon expulsion thresholds and efficiencies. These varying scenarios have been built on actual geochemical data and subsequently modelled by computer. Therefore, the envelop lines in this study are not randomly drawn but are guided by those well-established models. This has to been clearly explained because it is critical for the determination of the ASDLs.

Response: Thank you for your insightful suggestions. We agree with the reviewer that the model is well-established and supported by numerous successfully applied cases in Chinese basins. As the reviewer suggested, we have clarified the determination of those envelope lines in the revised manuscript and provided more references for the scientific community to review.

Changes: We have added a part of content in section 3.1 clarifying the determination of envelope lines, and the detailed revisions can be found at the end of this response, which is attached as a marked-up manuscript.

The references added are listed here:

1. Zhou, J., & Pang, X. Q.: A method for calculating the quantity of hydrocarbon generation and expulsion. Petroleum Exploration and Development, 29(1), 24-27, 2002.

2. Pang, X., Li, S., Jin, Z., & Bai, G.: Quantitative assessment of hydrocarbon expulsion of petroleum systems in the Niuzhuang sag, Bohai Bay Basin, East China. Acta Geologica Sinica - English Edition, 78(3), 615-625, doi:10.1111/j.1755-6724.2004.tb00174.x, 2004.

3. Jiang, F., Pang, X., Bai, J., Zhou, X., Li, J., & Guo, Y.: Comprehensive assessment of source rocks in the Bohai Sea area, eastern China. AAPG Bulletin, 100(6), 969-1002, doi.org/10.1306/02101613092, 2016.

4. Peng, J., Pang, X., Shi, H., Peng, H., & Xiao, S.: Hydrocarbon-generation potential of upper Eocene Enping Formation mudstones in the Huilu area, northern Pearl River Mouth Basin, South China Sea. AAPG Bulletin, 102(7), 1323-1342, doi:10.1306/0926171602417005, 2018.

**(2) Second, this study proposes a linear equation to predict ASDL using basin heat flow data. It looks pretty good with the data from the 6 basins in the study, but there are only 6 data points. Statistically, the prediction is not be that robust. Two other things may also cause inaccurate predictions using this equation. First, basin heat flow (HF) can vary significantly over the geologic history. This paper did not clearly state the heat flow is the paleo-HF or present HF of the basins. Second, I assume the depth is the present-day burial depth, which can be very different from the maximum burial depth in the basin history, especially foreland basins (e.g. Appalachian Basin, USA). The late-stage uplifting in the Appalachian Basin can be up to thousands of meters. Marcellus Shale in the northeast Pennsylvania, for example, shows very high thermal maturity (3 – 4.5% Ro) and in some area, has passed the ASDL, but the present-day depth generally is less than 3 – 4 km. In contrast, some rifted basins have never undergone any uplifting at all. Given the complex geologic histories of some basins, thermal maturity (Ro) appears to be more reliable (also pointed out in this paper). Although different types of basins are included in the study and the predicted ASDLs appear to be very reasonable, it would be nice to run some blind test including some data from basins in other continents. Also, it would be great to add some assumptions regarding the possible limitations of this application. In summary, this paper has a well-defined objective, a huge amount of data, a well**

proved methodology, and very solid results. The publication of this paper will bring a remarkable impact to the oil industry and the new findings from this paper surely will help lower exploration risks.

Response: Thank you for your insightful comments. The heat flow used in this study is the present heat flow value of a basin. As suggested by the reviewer, burial history could have a great impact on ASDL, and we have already discussed it in the manuscript and proposed similar conclusions that the depth corresponding to ASDL should be utilized carefully in the superimposed basin, and that thermal maturity is more suitable to characterize ASDL. On the other hand, the reviewer has also mentioned the blind test of our model in other basins. However, as we mentioned in the reply to the reviewer György Pogácsás, the equation proposed by our previous manuscript is only to demonstrate the existence of relationship among heat flow, organic matter type and the ASDL, instead of predicting the ASDL of a basin by the equation. This is because two other important factors have not been incorporated into the equation of which the establishment is limited by our database. Therefore, the construction of a complete model or equation still needs help from the scientific community to enrich the relevant database.

Changes: We have re-written section 3.3 according to the above response, and the detailed revisions can be found at the end of this response, which is attached as a marked-up manuscript.

*Short Comment #3 by Zhihong Pan*
General Comments: Fossil fuel resources play an important role in the modern life. As the increasing demand of hydrocarbon, how to streamline the exploration is of great significance. As is well known, the vertical distribution of hydrocarbon varies from several hundred meters to tens of kilometers, which can bring high risks to the oil and gas assessment and exploration. This paper introduces an innovative concept: Active Source Rock Depth Limits (ASDL), which defines the maximum burial depth for source rocks to generate hydrocarbon in sedimentary basins. The concept can significantly provide guidance for hydrocarbon assessment and exploration. Detailed examples from six major petroliferous basins in China have been presented. The method used to define ASDL is very solid, parameters applied in this paper like H/C ratios, "A"/TOC, "S1+S2"/TOC are widely used in organic geochemistry. The result shows the existence of ASDL, and it is comparable across these basins

regarding thermal maturity. Apart from the basins in China, the paper also shows the concept of ASDL is applicable to the basins all over the world, which strongly support the authors' idea. Moreover, the controlling factors of ASDL has been investigated in the paper and the authors point out the ASDL is mainly constrained by the heat flow of the basin and types of organic matter. Quantitative research in hydrocarbon geology has always been a challenging task, however, through geochemical and mathematical analysis, the quantitative relations between the above two controlling factors and ASDL is established in this paper, which is a good try. At the ending part of the paper, the authors come up with the idea that the petroliferous basin can be vertically divided into three parts based on HET, oil and gas supply limits, and each part has its own types of hydrocarbon reservoirs. The idea provides a more efficient way to target hydrocarbon reservoirs, which is very interesting and hopefully we can see more details about this in the future. Honestly speaking, I believe the work of this paper is very innovative and the result is very solid, and it provides us with a new perspective when doing fossil fuel exploration, which definitely deserves to be published. However, I still have few suggestions and comments for this paper:

(1) Geochemical data and hydrocarbon reservoir parameters from six representative petroliferous basins in China are quite enough. I am not sure whether the corresponding dataset for basins from worldwide is accessible, like Persian Gulf Basin, West Canada Basin, North Sea Basin. If one or few detailed examples of above mentioned basins can be presented, it would be the icing on the cake.

Response: Thanks for your comment. The datasets from global basins are mainly sourced from IHS (2010), which mainly contains datasets related to basins and hydrocarbon reservoirs rather than these geochemical data with source rocks.

Changes: According to the above response to the reviewer, there is no change in the revised manuscript related to this comment.

(2) In the part of quantitative prediction of ASDL, the predicted ASDL can be expressed by HI and HF, where HI is the proxy of organic matter types. As we all know, there are some other parameters can represent kerogen type, why the HI is selected to establish the quantitative relation in this paper?

Response: Thanks for your comment. It is true that many different methods could be taken to identify the organic matter type, and it is also feasible to convert the organic matter type parameters obtained by other methods into HI. However, there are two main reasons for selecting the hydrogen index (HI) as the input of our model. First, HI is a quantitative proxy for the characterization of kerogen types, and it can be easily obtained through Rock-Eval analysis. Numerous studies on source rock evaluation from the scientific community have proven the dependability and reliability of HI. Furthermore, HI has been widely chosen as the indicator of kerogen type in professional software such as PetroMod that is often utilized by the industrial community. Therefore, from the perspective of scientific and industrial community, HI is capable of serving as an input parameter to our model to indicate organic matter types.

Changes: According to the above response, we have added the reason in section 3.3 for using hydrogen index as the proxy of organic matter type, and the detailed revisions can be found at the end of this response, which is attached as a marked-up manuscript.

**(3) In part 3.1, the average thermal maturity level is regarded as the identification criterion for ASDL in general geological settings. However, in part 3.3, the quantitative prediction of ASDL is express by active source rock depth limit. As is known, the thermal evolution of organic matter is a chemical process and it is a function of time and heat. Therefore, I believe if we want express ASDL by depth, it is better to take the age of source rock into consideration when doing quantitative investigation.**

Response: Thanks for your comment. We agree with the reviewer that the quantitative model with the stratigraphic age taken into consideration is much more rigorous. However, as we mentioned in the reply to reviewer György Pogácsás, the ASDL is not only influenced by the heat flow and organic matter type, but also influenced by the stratigraphic age and tectonic uplift. To set up the equation with four independent variables, at least twelve equations with different variables should be applied. This is, however, not satisfied by our database since we only have datasets from six basins. It looks promising to construct a complete model or equation if more and more other basins' ASDLs are unravelled.

Changes: We have re-written section 3.3 according to the above response, and the detailed revisions can be found at the end of this response, which is attached as a

marked-up manuscript.

**(4) In part 3.3, it seems that the HF used here is the value of nowadays. But HF varies in geological history and can be influenced by tectonic events. For example, the current heat flow in Basin and Range is high due to the recent tectonic extension. Therefore, maybe an average heat flow from the time when the source rock deposited to present is a better.**

Response: Thanks for your comment. In our revised section 3.3, the equation only shows that there exists a relationship between heat flow and organic matter type and the ASDL, and the equation cannot be used to precisely predict the ASDL of a basin. We argue that it is better to predict the ASDL with the criteria provided by our study (i.e. Ro=3.5% ± 0.5%) through the basin modelling. We have tested the relationship between average paleo-heat flow and the ASDL, and an obvious negative relationship was observed, the same with the relationship between present average heat flow and the ASDL. Since the evolution processes of different basins are very complicated and the heat flow both in the present and in the geological past contribute to the thermal maturation of source rocks, it is therefore unnecessary to separate them if we only want to prove there is a relationship between heat flow and ASDL.

Changes: We have re-written section 3.3 according to the above response, and the detailed revisions can be found at the end of this response, which is attached as a marked-up manuscript.

**(5) In conclusion 1 and 2, what is meaning of ASRL? Is this a typo or Does it represent something else?**

Response: It should be ASDL rather than ASRL. We are sorry for the confusion caused by the typo, and we have modified it in a revised manuscript.

Changes: We have revised the ASRL into the ASDL.

***Short Comment #4 by Fengtao Guo***
**General Comments: I find this manuscript is generally well written, structured, and illustrated. Some of the results are very interesting and thought-provoking. It puts forward the concept of "Active Source Rock Depth Limits (ASDL)", and try to characterize the vertical depth distribution of discovered reservoirs. A**

huge of data has been systematically compiled around the world, especially the six key basins in China. The use of four methods to characterize and corroborate the ASDL, including the possibility to be used around the world, makes it more convincing. The controls on the ASDL are also explored and a quantitative model was established to predict the results. The study is very meaningful in the way that it is the first systematic attempt to work on the relationship of depths and hydrocarbon reservoirs, especially on such a huge scale around the world, with so much data. It brings this topic to our attention which should be studied before. The topic has great scientific values. It could help us to better understand why at a shallow depth there are no reservoirs in a basin whereas in some other basins, reservoirs are found in a much deeper layer. If proven, it will also help us to determine whether to drill a well to a certain depth. During my reading of this MS, I had several questions or suggestions for the authors. If they could help with them, I would really appreciate it.

(1) The ASDL was mostly established with the data of six basins in China, although data of basins around the world (IHS, 2010) were later used to verify the ASDL. Is it possible to incorporate some basins outside of China in the process of establishing the model? I totally understand this is totally a data issue and the authors may not be able to get enough systematic data of basin around the world as in China. However, this could be an improvement, if possible.

Response: Thanks for your suggestion. We agree with the reviewer that incorporating more basins into the model is much better for establishing a complete mode. As pointed out by the reviewer, however, we cannot get the geochemical data of source rocks like ours.

Changes: According to the above response to the reviewer, there is no change in the revised manuscript related to this comment.

(2) In section 3.3, the authors used the average values (heat value, depth, HI) of each basin to verify the model of ASDL, it is generally OK and understandable as there must be lots of data for each basin. However, is there still any possibility that some outliers may be present? If yes, how to explain them?

Response: Thanks for your comment. In the section 3.3, the HI value is given

according to source-rock general property, while the average heat flow of each basin is sourced from numerous literature, and the average depth of ASDL for each basin is obtained from various indicators as proposed in the manuscript. Therefore, the average heat flow and depth of ASDL for each basin may be the source of some outliers which could be caused by experimental errors. However, we have discarded the outliers during the compilation of these data.

Changes: According to the above response to the reviewer, there is no change in the revised manuscript related to this comment.

**(3) In section 3.4, the authors proposed the concept of Hydrocarbon Reservoir Depth Limit (HRDL), mentioning that "at some depth (Hydrocarbon Reservoir Depth Limit), the probability of drilling oil or gas reservoirs decreases to zero", and talks a bit about the relationship between HRDL and ASDL. I think this is an important part as for a hydrocarbon reservoir to form, it requires both hydrocarbons from source rocks and reservoirs rock to accumulate. Unfortunately, very little was discussed on the HRDL in this point. If possible, could more details be added on this?**

Response: Thanks for your suggestion. The HRDL is modified as HADL in the revised manuscript, and we have added relevant content in the revised manuscript. The HADL is a newly proposed concept by the first author, which is influenced by many different factors and we will discuss it in the other paper.

Changes: We have modified section 3.4 according to the above response, and the detailed revisions can be found at the end of this response, which is attached as a marked-up manuscript.
* * *

[revised manuscript text omitted]

---

## Referee Report (RR1)

General comment

Xiongqi Pang and his co-authors took into consideration the advices of the reviewers and practically rewrote most part of the earlier version (submitted on 06 May 20119) of their „Active Source Rock Depth Limit and its Controlling on the Formation and Occurrence of Fossil Fuel Resources" (essd-2019-72) paper. Their rewriting efforts strongly improved the manuscript.

Technical comments

Some minor technical corrections are listed as follows. I suppose besides the listed minor corrections an overall English language rechecking by a native English reviewer and/or editor might be useful.

Minor correction proposals

Page 7, line 7.  Instead of „Peters, 2018" „Peters et al 2018" is recommended

Page 10, line 6, instead of „lower" „shallower" is recommended

Page 12, line 9, instead of „china" „China" is recommended

Page 14, line 10, instead of „AAPG bulletin" „AAPG Bulletin" is recommended

Page 19, line 3, instead of „with" „versus" is recommended

Page 22, Figure 5 instead of „measured" ASDL a different expression is recommended (Sensu Stricto ASDL cannot be measured just estimated)

Page 27, Figure 10, instead of „Strong" Heat Flow „High" Heat Flow is recommended

---

## Author Response (AR2)

Dear Editor and Reviewer,

We are deeply grateful for the editorial board's consideration of our manuscript, and we honestly appreciate the comments from the reviewer which helps us improve the readability, quality, and rigour of the manuscript. We revised the manuscript again following the suggestions, and rechecked the manuscript thoroughly. Changes are marked in the marked-up manuscript. In this response, we clarified these changes that have been made to the manuscript. The comments are in **bold** type, and the changes are in normal type with blue colour.

**Referee Comment #1 by György Pogácsás**

General comment: Xiongqi Pang and his co-authors took into consideration the advices of the reviewers and practically rewrote most part of the earlier version (submitted on 06 May 20119) of their "Active Source Rock Depth Limit and its Controlling on the Formation and Occurrence of Fossil Fuel Resources" (essd-2019-72) paper. Their rewriting efforts strongly improved the manuscript.

Technical comments: Some minor technical corrections are listed as follows. I suppose besides the listed minor corrections an overall English language rechecking by a native English reviewer and/or editor might be useful.

(1) Page 7, line 7. Instead of "Peters, 2018" "Peters et al 2018" is recommended. Response: Thanks for your careful check. We have modified the "Peters, 2018" into "Peters et al., (2018)" in the revised manuscript. Changes can be seen in the attached marked-up manuscript.

**(2) Page 10, line 6. Instead of "lower" "shallower" is recommended.**

Response: Thanks for your careful check. We have modified the "lower" into "shallower" in the revised manuscript. Changes can be seen in the attached marked-up manuscript.

**(3) Page 12, line 9. Instead of "china" "China" is recommended.**

Response: Thanks for your careful check. We have modified the "china" into "China" in the revised manuscript. Changes can be seen in the attached marked-up manuscript.

**(4) Page 14, line 10. Instead of "AAPG bulletin" "AAPG Bulletin" is recommended.**

Response: Thanks for your careful check. We have modified the "AAPG bulletin" into "AAPG Bulletin" in the revised manuscript. Changes can be seen in the attached

marked-up manuscript.

**(5) Page 19, line 3. Instead of "with" "versus" is recommended.**

Response: Thanks for your careful check. We have modified the "with" into "with" in the revised manuscript. Changes can be seen in the attached marked-up manuscript.

**(6) Page 22, Figure 5 instead of "measured" ASDL a different expression is recommended (Sensu Stricto ASDL cannot be measured just estimated).**

Response: Thanks for your constructive suggestion and careful check. We have modified the "measured" into "estimated" in the Fig.5 in the revised manuscript. Changes can be seen in the attached marked-up manuscript.

**(7) Page 27, Figure 10, instead of "Strong" Heat Flow "High" Heat Flow is recommended.**

Response: Thanks for your constructive suggestion and careful check. We have modified the "Strong Heat Flow" into "High Heat Flow" in the Fig.10 in the revised manuscript. Changes can be seen in the attached marked-up manuscript.

Additional changes: We have rechecked the content of the manuscript thoroughly, and the language has been checked by a native English speaker again. In addition, the legend of Fig.7 has been changed slightly. The changes are marked in the revised manuscript. Please see the attached marked-up manuscript for detailed revisions.

**Active Source Rock Depth Limit and its Controlling on the Formation and Occurrence of Fossil Fuel Resources**

Xiongqi Pang1,2\*, Chengzao Jia1,3, Kun Zhang2,4\*\*, Maowen Li5, Youwei Wang2,6, Junwen Peng7, Boyuan Li2, and Junqing Chen1,2

[revised manuscript text omitted]
                                         | H/C                                              | 8970/3.5                                                     | 8350/3.2      | —             | —           | 5800/3.5           | 5280/3.6          | 7100/3.4                                |  |  |
| composition                                                      | O/C                                              | 9050/3.6                                                     | 8450/3.2      | _             | —           | 5740/3.4           | 5280/3.6          | 7130/3.4                                |  |  |
| The variation of residual                                        | "A"/TOC                                          | 9050/3.6                                                     | 7850/3.0      | 7540/3.6      | 6450/3.3    | 5560/3.1           | 5330/3.7          | 6963/3.4                                |  |  |
| hydrocarbon                                                      | "S 1 "/TOC                            | 9290/3.8                                                     | 7960/3.0      | 7780/3.8      | 6500/3.4    | 5490/3.2           | 5400/3.9          | 7070/3.5                                |  |  |
| The variation of                                                 | "S 1 +S 2 "/TOC            | 9300/3.8                                                     | 8200/3.0      | 7700/3.8      | 6600/3.4    | 5900/3.3           | 5400/3.9          | 7183/3.5                                |  |  |
| hydrocarbon generation and expulsion                             | Ve                                               | 9210/3.8                                                     | 8200/3.0      | 7660/3.7      | 6520/3.4    | 5700/3.3           | 5500/4.0          | 7115/3.5                                |  |  |
| The average values obtained from different methods in each basin |                                                  | 9145/3.7                                                     | 8168/3.1      | 7670/3.7      | 6518/3.4    | 5698/3.3           | 5348/3.8          | 7094/3.5                                |  |  |
| The data used for identifying ASDLs (sample number/well number)  |                                                  | 2063/79                                                      | 5353/351      | 460/27        | 1329/149    | 1193/69            | 3236/611          | Total:
13634/1286                    |  |  |

Table 2 Comparison of active source rock depth limits in the six petroliferous basins of China

**Figure 1:** Location of the six representative petroliferous basins and five coal-accumulation areas in China. The studied petroliferous basins, plotted on the China mainland, are pigmented with different colors according to their locations in China. The five coal-accumulation areas, bounded by large geological structural belts, are mapped according to Zhu (2011).